# Localizing Memorized Regions in Diffusion Models via Coordinate-Wise Curvature Differences

**Gwangho Kim** [1]   **Sungyoon Lee** [1]

## Abstract

Diffusion models can unintentionally memorize training samples, raising concerns about privacy and copyright. While recent methods can detect memorization, they often rely on global or model-specific signals and provide limited insight into where memorization appears within a generated image. We provide a geometric characterization of local memorization as a coordinate-wise variance collapse. However, such collapse can also arise from intrinsic data constraints rather than overfitting. To isolate overfitting-driven memorization, we propose curvature-difference methods that subtract the curvature of an underfitted baseline, either the unconditional model or a less-trained version of itself. We further derive a score-difference proxy that provides a geometric explanation for the widely used score-difference-based detection metric. Experiments on Stable Diffusion, evaluated against ground-truth memorization masks, show that our method outperforms the prior attention-based localization method. Code is available at https://github.com/Gwangho99/mem-curv-diff.

## 1. Introduction

Diffusion models (Sohl-Dickstein et al., 2015; Ho et al., 2020) have become the dominant paradigm for high-quality image generation. However, recent studies (Carlini et al., 2023; Somepalli et al., 2023a) show that these models can memorize and reproduce training samples, raising serious privacy and copyright concerns. Importantly, memorization need not occur at the level of entire images; even partial reuse of training content can suffice to violate privacy or copyright constraints (Somepalli et al., 2023a; Webster,

2023; Chen et al., 2025).

Most prior works characterize memorization using global signals that assign a single scalar value to each generated sample. Wen et al. (2024) introduced a score-difference-based detection metric that does not require retrieval over the training set, based on the intuition that memorized samples exhibit abnormally strong dependence on text conditioning. Recent works have sought to interpret such global detection signals through a geometric lens. Ross et al. (2025) proposed a geometric framework, linking memorization to regions of low Local Intrinsic Dimensionality (LID), and Jeon et al. (2025) further characterized memorization as sharpness of the learned probability landscape, interpreting the score-difference metric as a measure of the sharpness gap between conditional and unconditional models. These geometric quantities provide principled global characterizations of memorization as low dimensionality or high sharpness. However, they remain inherently global: they quantify how many degrees of freedom collapse, but provide no insight into where this collapse occurs within an image.

Motivated by this limitation, Chen et al. (2025) argued that memorization is often local, affecting only specific regions rather than entire images. They further emphasized that even partial replication of training data can pose privacy and copyright risks, while global image-level similarity scores are easily diluted by variations in non-memorized regions. Based on this insight, they proposed Bright Ending (BE), which relies on cross-attention maps to extract spatial memorization masks.

This reveals two fundamental gaps in existing approaches. First, regarding spatial localization, current geometric methods provide only global characterizations, while BE relies on model-specific internal signals. A principled and model-agnostic geometric method for spatially localizing memorization is still missing. Second, regarding the underlying mechanisms of existing metrics, while recent works characterize the widely used score-difference detection metric (Wen et al., 2024) as a sharpness gap, it remains fundamentally unclear why the gap itself is the decisive signal and why the unconditional model serves as the appropriate reference.

We bridge these gaps by extending the geometric frame-

[1]Department of Computer Science, Hanyang University, Seoul, South Korea. Correspondence to: Sungyoon Lee <sungyoonlee@hanyang.ac.kr>.

*Proceedings of the 43rd International Conference on Machine Learning*, Seoul, South Korea. PMLR 306, 2026. Copyright 2026 by the author(s).

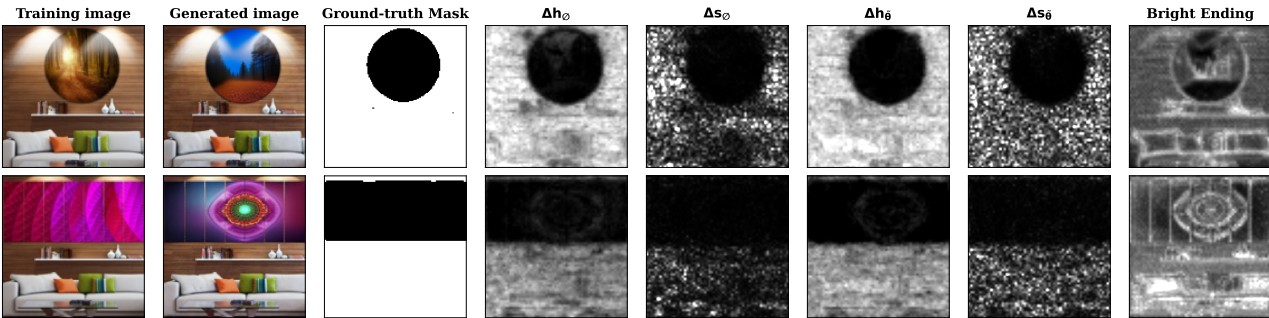

*Figure 1.* **Qualitative comparison of memorization localization.** For each example, we show the training image, the generated image, and the ground-truth memorization mask. We compare the proposed *curvature-difference-based* metrics $\Delta h_\varnothing$ (Eq. 1) and $\Delta h_{\tilde{\theta}}$ (Eq. 2), together with their *score-difference-based* surrogates $\Delta s_\varnothing$ (Eq. 5) and $\Delta s_{\tilde{\theta}}$ (Eq. 6), against the prior work, Bright Ending (BE) (Chen et al., 2025). Light regions indicate large values and hence high memorization scores.

work of Ross et al. (2025). Specifically, we characterize local memorization as a coordinate-wise variance collapse, which manifests geometrically as high curvature in the log-density. However, high curvature may also arise from intrinsically low-variance regions of the ground-truth data manifold and therefore does not, by itself, imply *undesirable* memorization. To isolate such effects, we introduce the curvature-difference-based method, which subtracts curvature estimated from an underfitted baseline—either an unconditional model or a less-trained conditional model. This removes intrinsic data structure driven curvature and highlights regions where variance collapses due to overfitting.

Finally, we show that the widely used score-difference-based metric (Wen et al., 2024) can be interpreted as an approximation to a curvature difference, providing a novel interpretation of existing heuristics and unifying them within our framework.

Our contributions can be summarized as follows:

- In Sections 4.1 and 4.2, we move beyond global metrics such as LID (Ross et al., 2025) and sharpness (Jeon et al., 2025), which only quantify the *total degrees of freedom*, and instead characterize memorization by revealing *where and along which directions* variance collapses via coordinate-wise analysis.

- In Section 4.3, we aim to localize memorization by removing curvature that arises from naturally low-variance regions of the data manifold, and introduce the curvature-difference-based method to isolate overfitting-driven memorization.

- In Section 4.4, we show that curvature-difference can be approximated using score differences, and further provide an alternative geometric interpretation of the existing score-difference-based metric (Wen et al., 2024) through this lens.

- In Section 5, we empirically demonstrate improved memorization localization on Stable Diffusion using ground-truth masks, outperforming the prior attention-based method (Chen et al., 2025).

## 2. Related Work

Early research focused on identifying memorization by exhaustively searching the entire training set. Carlini et al. (2023) employed a calibrated $\ell_2$ distance, while Somepalli et al. (2023a) used feature-space similarity metrics such as SSCD (Pizzi et al., 2022). However, these retrieval-based approaches are computationally prohibitive for large-scale models and datasets. To address scalability, Wen et al. (2024) introduced a score-difference-based detection metric that does not rely on external data retrieval, based on the intuition that memorized samples exhibit abnormally strong dependence on text conditioning. Ren et al. (2024) leveraged cross-attention patterns for memorization detection and mitigation.

Webster (2023) first distinguished *template verbatims*, which copy fixed spatial patterns with non-semantic variations, from *exact verbatims*, where the generated image is nearly identical to a training sample. Motivated by this distinction, Chen et al. (2025) explicitly argued for the need to analyze *local* memorization and proposed Bright Ending as a spatial localization mechanism based on cross-attention at the final denoising step. However, BE is inherently model-specific and, as shown in Figure 1, tends to produce false positives in non-memorized regions.

Several recent studies have attempted to interpret the score-difference-based metric (Wen et al., 2024) from a geometric perspective. Ross et al. (2025) proposed the Manifold Memorization Hypothesis (MMH), linking memorization to regions of low Local Intrinsic Dimensionality (LID). Jeon et al. (2025) further characterized memorization as sharpness in the probability landscape, arguing that Wen's metric

effectively measures the sharpness difference between conditional and unconditional models. While Jeon et al. (2025) interpret the metric as a sharpness gap, it remains unclear why the **gap itself** is the decisive signal and why the **unconditional model** is the appropriate reference.

We interpret the unconditional model as an underfitted baseline that captures the intrinsic curvature of the data manifold. From this viewpoint, the score difference can be understood as suppressing data-driven curvature and highlighting Overfitting-Driven Memorization (OD-Mem) (Ross et al., 2025), offering an alternative explanation for the effectiveness of score-difference-based detection. [1]

## 3. Preliminaries

### 3.1. Diffusion Models

Denoising diffusion models (Ho et al., 2020; Sohl-Dickstein et al., 2015; Song et al., 2021b) are a class of generative models that generate an image from $p_0(x)$ by gradually denoising data from pure Gaussian noise through a learned reverse diffusion process. At each timestep $t$, the forward process is given by:

$$q(x_t|x_0) = \mathcal{N}(x_t; \sqrt{\bar{\alpha}_t}x_0, (1 - \bar{\alpha}_t)I),$$

where $\alpha_t = 1 - \beta_t$ and $\bar{\alpha}_t = \prod_{s=1}^{t} \alpha_s$ and $\beta_t$ is the diffusion noise schedule at timestep $t$. In the continuous-time formulation (Song et al., 2021b), the forward diffusion process is defined by the forward stochastic differential equation (SDE):

$$dx_t = f(x_t, t) \, dt + g(t) \, dw_t,$$

where $w_t$ denotes standard Brownian motion. This process gradually perturbs data samples into Gaussian noise. Sampling is performed by simulating the reverse-time SDE

$$dx_t = \left[ f(x_t, t) - g(t)^2 \nabla_{x_t} \log p_t(x_t) \right] dt + g(t) \, d\bar{w}_t,$$

where $\bar{w}_t$ denotes Brownian motion in reverse time. In practice, $\nabla_{x_t} \log p_t(x_t)$ is replaced by its neural approximation $s_\theta(x_t, t)$.

### 3.2. Geometric Framework for Memorization

In this section, we review prior work (Ross et al., 2025) that characterizes memorization as a form of low dimensionality in the learned data manifold, which serves as the foundation for our approach.

Ross et al. (2025) introduced the *Manifold Memorization Hypothesis* (MMH), a geometric framework that explains memorization in deep generative models through the lens of the manifold hypothesis (Bengio et al., 2013; Loaiza-Ganem

et al., 2024). The MMH posits that memorization occurs at a point $x$ when the manifold learned by the model, $\mathcal{M}_\theta$, has an intrinsic dimensionality that is significantly lower than expected or fails to match the ground truth manifold $\mathcal{M}_*$.

The framework formalizes this using the *Local Intrinsic Dimension* (LID), denoted as $LID(x)$, which represents the number of degrees of freedom or valid independent directions of movement at a specific point $x$. Based on the relationship between the ground truth $LID_*$ and the model's learned $LID_\theta$, Ross et al. (2025) categorized memorization into two distinct types:

- **Overfitting-driven Memorization (OD-Mem):** This is characterized as a modeling failure where the model fails to generalize correctly to the ground-truth distribution ($LID_\theta(x) < LID_*(x)$). In this scenario, $\mathcal{M}_\theta$ is too constrained compared to the idealized ground truth manifold $\mathcal{M}_*$, meaning the model lacks the necessary degrees of freedom to capture the full complexity of the ground-truth data manifold.

- **Data-driven Memorization (DD-Mem):** This occurs when the model successfully captures the ground truth distribution ($LID_\theta(x) \approx LID_*(x)$), but the ground truth manifold itself possesses a small local intrinsic dimension ($LID_*$) at the given point. Rather than being a modeling failure, DD-Mem is an inherent consequence of the properties of $p_*(x)$, such as low data complexity that significantly reduces the available degrees of freedom within the data manifold.

However, we do not treat DD-Mem as undesirable memorization, limiting its definition to the intrinsic low-variance structure of the data manifold, as we detail in Section 4.3. While Ross et al. (2025) include data duplication under DD-Mem, we exclude such sampling-bias-induced regularity from our definition of intrinsic data properties.

**Notations.** Let $x_t \in \mathbb{R}^d$ denote the diffusion variable at timestep $t$. We denote the unconditional and conditional model distributions by $p_\theta(x_t)$ and $p_\theta(x_t \mid c)$, respectively. The neural network is trained to approximate the corresponding score functions,

$$s_\theta(x_t) \approx \nabla_{x_t} \log p(x_t), \quad s_\theta(x_t, c) \approx \nabla_{x_t} \log p(x_t \mid c),$$

via denoising score matching.

We define the Hessian of the log-density induced by the model as

$$H_\theta(x_t) := \nabla_{x_t} s_\theta(x_t), \quad H_\theta(x_t, c) := \nabla_{x_t} s_\theta(x_t, c),$$

which serves as a second-order approximation to the curvature of $\log p_\theta$.

---

[1]See Appendix B for extended related work on memorization mitigation.

# 4. Localizing Memorized Regions via Curvature Difference

In this section, we present a geometric framework to explicitly localize memorized regions within generated samples. Section 4.1 characterizes local memorization as a coordinate-wise collapse of variability, distinguishing it from global dimensionality measures. Section 4.2 establishes the theoretical link between this variance collapse and the curvature of the log-density. Section 4.3 introduces a difference-based metric that subtracts the curvature of an underfitted baseline to isolate overfitting-driven memorization from intrinsic data constraints. Finally, Section 4.4 derives a score-difference-based proxy for this curvature metric, offering a novel geometric interpretation of the widely used detection metric (Wen et al., 2024).

## 4.1. Coordinate-Wise Variance and Memorization

As discussed in Section 3.2, memorized samples, often characterized by low local intrinsic dimensionality (LID), are commonly understood as having fewer effective degrees of freedom than non-memorized samples (Ross et al., 2025). While such global geometric quantities provide useful signals, we argue that they are insufficient to fully characterize local memorization (Chen et al., 2025) or template verbatim (Webster, 2023). In particular, they fail to capture how the available degrees of freedom are distributed across the data space—a limitation we address by characterizing memorization through coordinate-wise variance.

To illustrate this limitation, consider a synthetic example in $\mathbb{R}^4$ where two different distributions share the same intrinsic dimensionality (LID $= 2$), as shown in Figure 2. In Figure 2a, variability is distributed across all coordinates, resulting in relatively uniform *coordinate-wise* covariance. In contrast, in Figure 2b, the same two degrees of freedom are concentrated on a specific subset of coordinates, while the remaining coordinates are fixed. Although both distributions have identical intrinsic dimensionality, the latter leaves the two pixels fixed, providing a strong signal of memorization for those specific pixels.

This distinction naturally scales to high-dimensional image spaces. Consider a generative model producing $64 \times 64$ images. Two samples, $x, y \in \mathbb{R}^{4096}$, may both have LID $= 64$, yet differ fundamentally in how variability is expressed:

- For sample $x$, the 64 degrees of freedom are distributed globally across the image, manifesting as coherent changes in object shape, pose, or illumination.

- For sample $y$, the same 64 degrees of freedom are localized within a single $8 \times 8$ patch, while the remaining $4096 - 64$ pixels are effectively fixed and closely match a specific region of some training images.

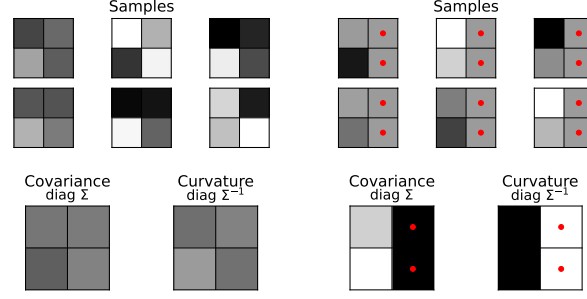

*(a)* concept memorization        *(b)* verbatim memorization

*Figure 2.* Samples and coordinate-wise curvature of linear Gaussian models $x = Az + \varepsilon$ with $z \sim \mathcal{N}(0, I_2)$ and $\varepsilon \sim \mathcal{N}(0, \sigma^2 I_4)$ ($\sigma \ll 1$). Both constructions have $\mathrm{rank}(A) = 2$ and identical Frobenius norm $\|A\|_F$, and thus share the same intrinsic dimensionality. **Top**: samples reshaped as $2 \times 2$ images. **Bottom**: coordinate-wise curvature $-\mathrm{diag}(\nabla_x^2 \log p(x)) = \mathrm{diag}((AA^\top + \sigma^2 I)^{-1})$. In **(a)**, $A$ has dense rows so that variability is distributed across all coordinate directions, whereas in **(b)**, $A$ contains two zero rows so that $x$ has near-zero variance in the two coordinate directions. We mark the near-zero variance coordinates in red. Light colors indicate higher values.

Despite identical global intrinsic dimensionality, these samples have qualitatively different semantic interpretations. The former reflects meaningful, concept-level variability, whereas the latter corresponds to a template-based reproduction with limited spatial support. This difference cannot be captured by global measures such as LID or sharpness alone, as they quantify the *number* of degrees of freedom but not *where* those degrees of freedom are expressed.

We refer to the latter phenomenon as *verbatim memorization*—encompassing both exact and template verbatims following Webster (2023)—and distinguish it from the former, which we term *concept memorization*. This distinction highlights a fundamental limitation of existing geometric characterizations: while they effectively measure overall dimensionality or sensitivity, they are inherently blind to both *where* memorization occurs and *what* is memorized.

The primary focus of this paper is to characterize verbatim memorization, specifically *template verbatim* (or equivalently, *local memorization*), by explicitly identifying where memorization occurs within a generated sample, that is, which coordinates exhibit a collapse of variability.

## 4.2. Linking Variance to Curvature

In the previous section, we argued that local memorization is characterized by a coordinate-wise collapse of variability rather than a global reduction in degrees of freedom. Motivated by the geometric interpretation of sharpness and curvature in Jeon et al. (2025), we now link this phenomenon to the curvature of the learned data distribution.

As illustrated in Figure 2, for Gaussian distributions, the

curvature of the log-density, given by $-\nabla_x^2 \log p(x)$, is directly related to the inverse covariance matrix. Coordinates with small variance correspond to directions of high curvature, whereas directions with large variability exhibit relatively flat curvature. This observation naturally bridges the covariance-based intuition developed in the previous section with a curvature-based characterization of local memorization. Furthermore, in diffusion models, the covariance structure of the final sample $x_0$ is reflected in the curvature of the model distribution at intermediate noisy states $x_t$. This relationship is formalized by the following proposition.

**Proposition 4.1.** *Let the forward transition kernel be*

$$p(x_t \mid x_0) = \mathcal{N}\left(\alpha_t\, x_0,\; \sigma_t^2 I\right),$$

*and define the marginal*

$$p(x_t) = \int p(x_t \mid x_0)\, p(x_0)\, dx_0.$$

*Then the conditional covariance satisfies*

$$\mathrm{Cov}[x_0 \mid x_t] = \frac{1}{\alpha_t^2}\left(\sigma_t^4 \nabla_{x_t}^2 \log p(x_t) + \sigma_t^2 I\right).$$

See Appendix D.2 for a proof. This proposition shows that large values of $\left(-\nabla_{x_t}^2 \log p(x_t)\right)_{ii}$ correspond to low conditional variance along coordinate $i$, indicating a collapse of variability and hence signals local memorization in that direction. Motivated by this result, we utilize the diagonal of the Hessian, $\mathrm{diag}(-H_\theta(x_t, c))$, as a metric to capture coordinate-wise variance collapse.

### 4.3. Isolating Overfitting-Driven Memorization

While the coordinate-wise curvature, $\mathrm{diag}(-H_\theta(x_t, c))$, provides a natural extension of the Manifold Memorization Hypothesis (MMH), we argue that this metric alone is insufficient for isolating undesirable memorization. As shown in the second column of Figure 3, $\mathrm{diag}(-H_\theta(x_t, c))$ often exhibits high curvature in regions that are intrinsically constrained by the data distribution, reflecting generic structural simplicity rather than memorization.

More specifically, when a prompt explicitly specifies "a black background" (as in the "Non Mem" case in Figure 3), the corresponding background pixels are naturally forced to take near-constant values across generations. This lack of variability is not the result of overfitting to a particular training instance, but rather a direct consequence of semantic constraints imposed by the prompt and the underlying data manifold. Even in the "Local Mem" case, non-memorized regions may still exhibit high curvature due to their inherent data simplicity, as observed in generated samples.

While such behavior corresponds to *data-driven memorization (DD-Mem)* within the MMH framework, treating DD-

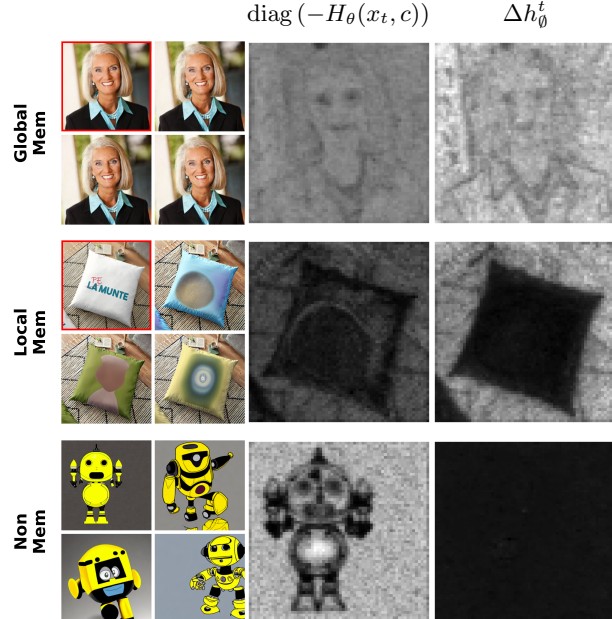

$$\mathrm{diag}\left(-H_\theta(x_t, c)\right) \qquad \Delta h_\emptyset^t$$

*Figure 3.* **Curvature-difference isolates overfitting-driven memorization.** Heatmaps within each column share the same color scale, with light and dark regions indicating high and low curvature (or differences), respectively. **Left:** Generated samples; training instances are highlighted with red borders. **Middle:** Coordinate-wise curvature $\mathrm{diag}(-H_\theta(x_t, c))$ computed at the final sampling step ($t \approx 0$) of the first generated sample in the top row. **Right:** Proposed curvature-difference $\mathrm{diag}(-H_\theta(x_t, c)) - \mathrm{diag}(-H_\theta(x_t))$, evaluated in the same way as Middle. By subtracting the unconditional baseline, this metric suppresses data-driven curvature and selectively highlights overfitting-driven memorization, yielding sharper spatial localization.

Mem as memorization—at least for the purpose of localizing unintended memorization—is conceptually misleading. High curvature induced by intrinsic data constraints does not indicate that the model has collapsed onto a specific training example, but instead reflects expected structure of the data distribution. In this work, we therefore explicitly exclude such intrinsic constraints and focus exclusively on isolating *overfitting-driven memorization (OD-Mem)*, where excessive curvature arises from abnormal variance collapse induced by overfitting. To achieve this, we propose subtracting the curvature of a relatively *underfitted* baseline model from that of the original model.

We first consider the unconditional model as a baseline. However, as recently analyzed by Karras et al. (2024), the unconditional denoiser can be interpreted as a relatively underfitted model, not merely due to the absence of conditioning, but because it must represent the entire data distribution at once, whereas the conditional model only needs to fit a single class or prompt for each sample. This mismatch in task complexity leads the unconditional model to attain a looser fit to the data. Motivated by this view, we further

consider a less-trained version of the model as an alternative underfitted baseline. We define the two metrics based on curvature difference as:

$$\Delta h_\emptyset^t := \mathrm{diag}(-H_\theta(x_t, c)) - \mathrm{diag}(-H_\theta(x_t)), \quad (1)$$

$$\Delta h_{\tilde{\theta}}^t := \mathrm{diag}(-H_\theta(x_t, c)) - \mathrm{diag}(-H_{\tilde{\theta}}(x_t, c)), \quad (2)$$

where $\tilde{\theta}$ refers to a less-trained version of the model that captures the general data manifold while being relatively less overfitted to specific training instances. [2]

By removing the baseline curvature induced by intrinsic data simplicity, this metric highlights regions where the $\mathrm{diag}(-H_\theta(x_t, c))$ significantly exceeds what is expected from the data manifold alone. As demonstrated in the third column of Figure 3, this approach yields substantially clearer and more precise localization of memorized content compared to raw curvature alone.

In practice, explicitly forming the full Hessian is intractable in high-dimensional settings. Instead, we efficiently approximate the diagonal of the Hessian differences using the Hutchinson estimator (Hutchinson, 1989). This estimator requires only Hessian–vector products, which can be computed efficiently via automatic differentiation without explicitly forming the full Hessian. Details are provided in Appendix D.1.

### 4.4. A Novel Geometric Interpretation of Wen's Metric

We provide a link between the curvature-difference framework introduced in the previous section and *score difference* by exploiting a Fisher-information-type identity (Lehmann & Casella, 1998). Through this connection, we offer a novel geometric interpretation of the widely used score-difference-based metric proposed by Wen et al. (2024):

$$\|s_\theta(x_t, c) - s_\theta(x_t)\|_2. \quad (3)$$

Also, while Hutchinson-based estimators already provide an efficient approach to estimating curvature, utilizing score differences as a proxy further improves efficiency by entirely avoiding Hessian computations. Note that our primary interest is $\nabla_{x_t} \log p(x_t|c) - \nabla_{x_t} \log p(x_t) = \nabla_{x_t} \log p(c|x_t)$.

**Proposition 4.2.** *Let $p(c \mid x)$ be a conditional likelihood that is twice continuously differentiable with respect to $x$. Define the Fisher information matrix with respect to $x$ as*

$$\mathcal{I}(x) := \mathbb{E}_{c \sim p(c|x)}\big[\nabla_x \log p(c \mid x) \nabla_x \log p(c \mid x)^\top\big].$$

*Then $\mathcal{I}(x)$ satisfies the Fisher information identity:*

$$\mathcal{I}(x) = \mathbb{E}_{c \sim p(c|x)}\big[-\nabla_x^2 \log p(c \mid x)\big]. \quad (4)$$

*Moreover, by taking the diagonal terms:*

$$\mathbb{E}_{c \sim p(c|x)}\big[\mathrm{diag}\big(-\nabla_x^2 \log p(x|c) + \nabla_x^2 \log p(x)\big)\big]$$
$$= \mathbb{E}_{c \sim p(c|x)}\big[\big(\nabla_x \log p(x|c) - \nabla_x \log p(x)\big)^{\odot 2}\big].$$

*where $\odot 2$ denotes element-wise squaring.*

See Appendix D.3 for a proof. Based on this proposition, we define the squared score difference

$$\Delta s_\emptyset^t := (s_\theta(x_t, c) - s_\theta(x_t))^{\odot 2} \quad (5)$$

as a computationally efficient proxy for the curvature difference $\Delta h_\emptyset^t$. Crucially, this proxy becomes increasingly reliable at late sampling steps. As $t \to 0$, $x_t$ becomes increasingly informative about the specific condition under which it was generated. Consequently, replacing the expectation in Eq. 4 with the generating condition becomes more accurate, making the score difference a faithful proxy of the curvature difference in the final stages of generation.

Our derivation reveals that Wen's metric (Eq. 3) is essentially a spatially aggregated instance of our coordinate-wise curvature difference principle. Prior works primarily attributed the efficacy of Eq. 3 to the observation that memorized prompts exhibit *abnormally strong text conditioning*, essentially interpreting the metric as a measure of guidance strength (Wen et al., 2024; Chen et al., 2025). In contrast, our framework highlights the crucial role of the unconditional term as a geometric baseline. We interpret the unconditional model as a *relatively underfitted baseline* that captures the intrinsic curvature of the data manifold. From this perspective, the subtraction in Eq. 3 does not merely measure conditioning strength, but rather filters out the intrinsic data complexity shared by both models. This effectively isolates the *overfitting-driven curvature* induced purely by memorization, providing a geometric justification for the metric.

Finally, an analogous interpretation applies to the less-trained baseline $\tilde{\theta}$. We have $\nabla_{x_t} \log p(\theta|x_t, c) = \nabla_{x_t} \log p(x_t|\theta, c) - \nabla_{x_t} \log p(x_t|c)$. Here, the first term is the score of the fully trained target model, $s_\theta(x_t, c)$, and the second term is the marginal score representing the general data manifold, which we replace with the less-trained baseline $s_{\tilde{\theta}}(x_t, c)$. Applying Proposition 4.2 to $p(\theta|x_t, c)$ motivates the following surrogate:

$$\Delta s_{\tilde{\theta}}^t := (s_\theta(x_t, c) - s_{\tilde{\theta}}(x_t, c))^{\odot 2}. \quad (6)$$

Intuitively, this score difference captures the additional gradient signal acquired during late-stage training; by filtering out the general data manifold captured by the less-trained baseline, it serves as a computationally efficient proxy for the curvature induced by overfitting.

The overall procedure for computing our four metrics $(\Delta h_\emptyset, \Delta h_{\tilde{\theta}}, \Delta s_\emptyset, \Delta s_{\tilde{\theta}})$ is summarized in Algorithm 1.

---

[2] Note that $\Delta h_\emptyset$ offers a more practical implementation using the unconditional model already utilized for CFG; however, we also utilize $\Delta h_{\tilde{\theta}}$ as an alternative to further justify our framework.

*Table 1.* Quantitative evaluation of memorization localization. We report Intersection-over-Union (IoU) and Pixel Accuracy (ACC). While 'TV only' focuses on Template Verbatims, the 'All' setting evaluates the metrics on the complete dataset, which includes Template Verbatims, Global Memorization cases, and Non-memorized samples. For SD v2.1, we report 'TV + Non-mem' due to the absence of Global Memorization cases in the dataset.

| | SD v1.4 | | | | SD v2.1 | | | |
| | TV only | | All | | TV only | | TV + Non-mem | |
| Method | IoU | ACC | IoU | ACC | IoU | ACC | IoU | ACC |
|---|---|---|---|---|---|---|---|---|
| All-ones | 0.560 | 0.560 | 0.522 | 0.522 | 0.649 | 0.649 | 0.325 | 0.325 |
| All-zeros | 0.000 | 0.440 | 0.333 | 0.478 | 0.000 | 0.351 | 0.500 | 0.675 |
| BE (Chen et al., 2025) | 0.751 | 0.805 | 0.564 | 0.849 | 0.933 | 0.957 | **0.956** | 0.978 |
| $\mathrm{diag}(-H_\theta(x_t, c))$ | 0.586 | 0.600 | 0.522 | 0.696 | 0.649 | 0.649 | 0.500 | 0.675 |
| $\Delta h_\emptyset$ (Eq. 1) | 0.899 | 0.940 | **0.953** | **0.968** | 0.943 | 0.964 | 0.866 | 0.980 |
| $\Delta s_\emptyset$ (Eq. 5) | 0.830 | 0.896 | 0.918 | 0.951 | 0.785 | 0.840 | 0.794 | 0.919 |
| $\Delta h_{\tilde{\theta}}$ (Eq. 2) | **0.921** | **0.952** | 0.867 | 0.966 | **0.947** | **0.967** | 0.828 | **0.982** |
| $\Delta s_{\tilde{\theta}}$ (Eq. 6) | 0.863 | 0.917 | 0.654 | 0.904 | 0.920 | 0.947 | 0.844 | 0.973 |

## 5. Experiments

In this section, we present both qualitative and quantitative evaluations of the proposed metrics and compare them with the prior approach, Bright Ending (BE) (Chen et al., 2025), as well as the raw curvature $\mathrm{diag}(-H_\theta(x_t, c))$.

**Setup.** To evaluate our methods, we use Stable Diffusion (SD) v1.4 and v2.1 (Rombach et al., 2022). We also use SD v1.1 and SD v2.0 as less-trained versions of each model. Following standard practice in memorization detection studies, we generate samples using the DDIM sampler with 50 inference steps and classifier-free guidance with guidance scale 7.5 for all experiments (Song et al., 2021a; Ho & Salimans, 2022). We evaluate the four proposed metrics, $\Delta h_\emptyset^t$, $\Delta h_{\tilde{\theta}}^t$, $\Delta s_\emptyset^t$, and $\Delta s_{\tilde{\theta}}^t$, at the final sampling step, and use $K = 16$ Hutchinson samples. For notational simplicity, we omit the timestep superscript $t$ unless otherwise specified. For each metric, we aggregate values across the channel dimension by summation, yielding a single spatial map used for all qualitative and quantitative evaluations. More details are provided in Appendix F.

### 5.1. Evaluation Using Ground-Truth Masks

As noted in the limitations of Chen et al. (2025), their work primarily evaluates localization methods through *extrinsic* criteria, such as improvements in memorization detection performance. While such evaluations demonstrate practical utility, they do not directly assess whether the predicted masks faithfully capture the spatial structure of local memorization itself. In this section, we therefore perform a direct, spatial evaluation against ground-truth memorization masks.

**Construction of ground-truth masks.** We adopt the ground-truth template masks provided by Webster (2023), which are constructed by identifying spatial regions that remain invariant across near-duplicate training samples. We utilize the prompt–seed–template tuples identified by their pipeline, where images generated from potentially memorized prompts are matched to a template if their difference within the invariant regions—measured by masked mean squared error—falls below a strict threshold. This tuple-based identification allows us to strictly pair each generated sample with its corresponding template mask during evaluation. We categorize the ground-truth masks into three scenarios: (1) for template verbatims (TV), we use the specific invariant region masks defined by the matched template; (2) for global memorization (including matching and retrieved verbatims), we use all-ones masks; and (3) for non-memorized samples, we use all-zeros masks.

**Evaluation.** We generate metric maps for our proposed methods and the prior method Bright Ending (BE) (Chen et al., 2025), resizing them to $256 \times 256$ to match the resolution of the ground-truth masks. To ensure consistent thresholding across samples, we perform a global normalization to $[0, 1]$ independently for each metric over all evaluation samples. We then report the best Intersection-over-Union (IoU) and Pixel Accuracy (ACC) obtained by sweeping a uniform threshold over the $[0, 1]$ range, following standard segmentation evaluations (Kong et al., 2024; Tang et al., 2023). These IoU and pixel accuracy values are computed per sample and averaged over the evaluation set.

**Qualitative Results.** Qualitative comparisons are shown in Figure 1, with additional examples in Figures 4, and 5 in Appendix A. Across a wide range of examples, our metrics consistently produce activation patterns that are well aligned with visually identifiable memorized regions. In contrast, BE frequently exhibits bright activations in non-memorized regions across multiple prompts and seeds. Overall, although both approaches are able to highlight memorized regions, our metric exhibits more consistent qualitative align-

ment with the ground-truth masks, whereas BE tends to over-activate in non-memorized areas.

**Quantitative Results.** Table 1 summarizes the quantitative evaluation of localization accuracy using IoU and ACC against the ground-truth memorization masks. Compared to the raw curvature $\mathrm{diag}(-H_\theta(x_t, c))$, the proposed curvature-difference metrics and their approximated versions achieve substantially higher IoU and ACC across all settings. This improvement demonstrates that removing intrinsic data-driven curvature is essential for accurate localization. Additional results in Appendix G demonstrate similar efficacy on Realistic Vision v5.1.

The relatively strong performance of Bright Ending (BE) on SD v2 can be partially attributed to dataset characteristics. Approximately 85% of the memorized prompts in SD v2 begin with "Shaw Floors", which often produce images with visually simple floor regions, as illustrated in the first four rows of Figure 5. BE tends to be effective primarily when the non-memorized regions are visually simple. In contrast, when the images contain complex structures or textures, BE frequently exhibits bright activations even in non-memorized regions, as shown in rows 5–9 of Figure 5, indicating persistent false positives. This aligns with the observation by Chen et al. (2024) that the end token attention map tends to exhibit high values in foreground regions and low values in the background.

We also observe that the IoU of $\Delta s_{\tilde\theta}$ degrades in the SD v1.4 'All' setting, whereas it performs well in the 'TV-only' setting. To diagnose this effect, we apply the TV-optimal threshold to non-memorized samples and find that nearly 90% of them contain at least one outlier pixel whose activation exceeds the threshold, causing zero IoU for each non-memorized sample. Compared to curvature-based metrics, such outlier activations occur more frequently for $\Delta s_{\tilde\theta}$, reflecting the fact that score-based quantities provide a noisier approximation to the underlying curvature. Crucially, this issue is easily mitigated: applying a simple $13 \times 13$ mean filter suffices to suppress these sparse outliers, **improving the IoU to 0.820 and the ACC to 0.931**.

**Remark.** It is worth noting that SD v2.0, used here as the less-trained baseline ($\tilde\theta$), is already known to exhibit memorization (Webster, 2023). Nevertheless, our metric effectively localizes memorized regions in SD v2.1. This implies that the probability landscape continues to sharpen (i.e., curvature increases) as the model undergoes further training, even after the initial onset of memorization. Further discussion on this phenomenon is provided in Appendix E.

*Table 2.* Memorization detection performance. The detection score is defined as the spatial expectation ($\mathbb{E}[\cdot]$) of the localization map, averaged across 4 random seeds per prompt. Results are reported as AUC / TPR@1%FPR

| Method | SD v1.4 | SD v2.1 |
|---|---|---|
| $\mathbb{E}[\text{BE-attention}]$ | 0.886 / 0.390 | 0.945 / 0.877 |
| $\mathbb{E}[\mathrm{diag}(-H_\theta(x_t, c))]$ | 0.861 / 0.082 | 0.775 / 0.000 |
| $\mathbb{E}[\Delta h_\emptyset]$ | 0.997 / 0.982 | 0.995 / 0.950 |
| $\mathbb{E}[\Delta h_{\tilde\theta}]$ | 0.989 / 0.900 | 0.996 / 0.963 |
| $\mathbb{E}[\Delta s_\emptyset]$ | 0.997 / 0.982 | **0.997** / **0.968** |
| $\mathbb{E}[\Delta s_{\tilde\theta}]$ | **0.998** / **0.988** | 0.993 / **0.968** |

### 5.2. Memorization Detection

If a metric accurately localizes memorized regions within a generated sample, aggregating it over spatial coordinates should naturally yield a strong signal for memorization detection. We therefore evaluate whether our localization metrics also serve as effective detection metrics when summed over spatial dimensions.

**Detection Setup.** We compare our four difference-based metrics with the raw curvature baseline $-H_\theta(x_t, c)$, and additionally report a purely attention-based baseline obtained by aggregating the BE attention map (Chen et al., 2025). For each generated sample, we aggregate the corresponding localization map by averaging over all spatial positions and use the resulting scalar as a detection score. Following prior works, we evaluate detection performance using 500 memorized prompts from SD v1 and 219 prompts for SD v2 identified by Webster (2023) and non-memorized prompts used in Jeon et al. (2025), which are sourced from COCO (Lin et al., 2014), Lexica (Lexica, 2024), Tuxemon (HuggingFace, 2024), and GPT-4 (Achiam et al., 2023).

**Results.** The results are summarized in Table 2. Across all experimental settings, the four difference-based metrics consistently outperform the raw curvature baseline, $\mathbb{E}[-H_\theta(x_t, c)]$. This demonstrates that subtracting an underfitted baseline is essential for reliable detection. Furthermore, the superior performance of our method compared to BE-attention suggests that our proposed metrics align more closely with the underlying mechanics of memorization.

Notably, the aggregated score-difference metrics yield slightly higher detection accuracy than the curvature-based metrics. Although score estimates exhibited noisier behavior at the coordinate level—as observed in the localization experiments—spatial averaging appears to effectively cancel out local variance. This allows the strong global signal inherent in the score difference to serve as a robust detector.

We also observe that the score difference using a less-trained

baseline, $\mathbb{E}[\Delta s_{\tilde{\theta}}]$, achieves detection performance comparable to that of the unconditional baseline, $\mathbb{E}[\Delta s_{\emptyset}]$. This empirical result supports the geometric interpretation presented in Section 4.4. Specifically, it validates that the effectiveness of score-difference metrics stems from subtracting an underfitted baseline. This operation filters out intrinsic data complexity, thereby isolating the curvature induced specifically by overfitting.

## 6. Limitations and Future Work

Our first limitation is that the curvature-based metrics ($\Delta h_{\emptyset}$, $\Delta h_{\tilde{\theta}}$) are more time-consuming than the prior work (Chen et al., 2025), as they require Hessian–vector products. However, as shown in Tables 6 and 8 in Appendix H, even a single Hutchinson sample ($K = 1$) already achieves competitive localization accuracy, with diminishing returns for larger $K$. Moreover, we introduce score-difference surrogates ($\Delta s_{\emptyset}$, $\Delta s_{\tilde{\theta}}$) that approximate curvature differences without explicitly computing Hessians.

Second, our formulation is explicitly designed to capture *verbatim* memorization, characterized by a collapse of variability on a subset of spatial coordinates. As discussed in Section 4.1, samples that exhibit *concept-level* memorization (e.g., celebrities or artistic styles) may share the same global intrinsic dimensionality while distributing their degrees of freedom across the entire image. Such cases do not induce strongly localized curvature and are therefore not well captured by our metric. Extending our framework to distinguish and analyze concept-level memorization remains an important direction for future work.

## 7. Conclusion

In this work, we address two key gaps in diffusion model memorization research: the lack of an accurate, model-agnostic method for localizing memorized regions, and the limited understanding of why score or sharpness gaps—particularly relative to unconditional models—serve as effective memorization signals. We show that local memorization can be understood as coordinate-wise variance collapse, and that isolating overfitting-driven memorization requires subtracting intrinsic data-driven curvature using an underfitted baseline such as an unconditional or less-trained model. Building on this insight, we propose curvature-difference and score-difference frameworks that provide both principled spatial localization and a unified geometric explanation for existing memorization metrics.

## Impact Statement

This work enables more fine-grained analysis of memorization in diffusion models by localizing where memorization occurs within generated images. Such spatially resolved analysis can help better understand and diagnose privacy and copyright risks arising from partial reproduction of training data. By moving beyond global memorization scores, our approach provides more informative signals for analyzing memorization behavior in generative models.

## Acknowledgments

We thank the anonymous reviewers for insightful reviews. This work was partially supported by Institute of Information & communications Technology Planning & Evaluation (IITP) grants (RS-2020-II201373, Artificial Intelligence Graduate School Program (Hanyang University); RS-2023-002206284, Artificial intelligence for prediction of structure-based protein interaction reflecting physicochemical principles); the BK21 FOUR (Fostering Outstanding Universities for Research) project; NRF2024S1A5C3A02043653, Socio-Technological Solutions for Bridging the AI Divide: A Blockchain and Federated Learning-Based AI Training Data Platform) and Korea Institute for Advanced Study (KIAS) grant funded by the Korean government (MSIT).

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

# A. Additional Qualitative Results

We provide additional qualitative results and visualization details for Figures 1, 4, and 5. For each column, heatmaps are visualized using a shared color scale, where the minimum is set to the column-wise minimum value and the maximum is clipped at the 99th percentile. Additionally, for the curvature-difference metrics ($\Delta h$), negative values are clipped to zero.

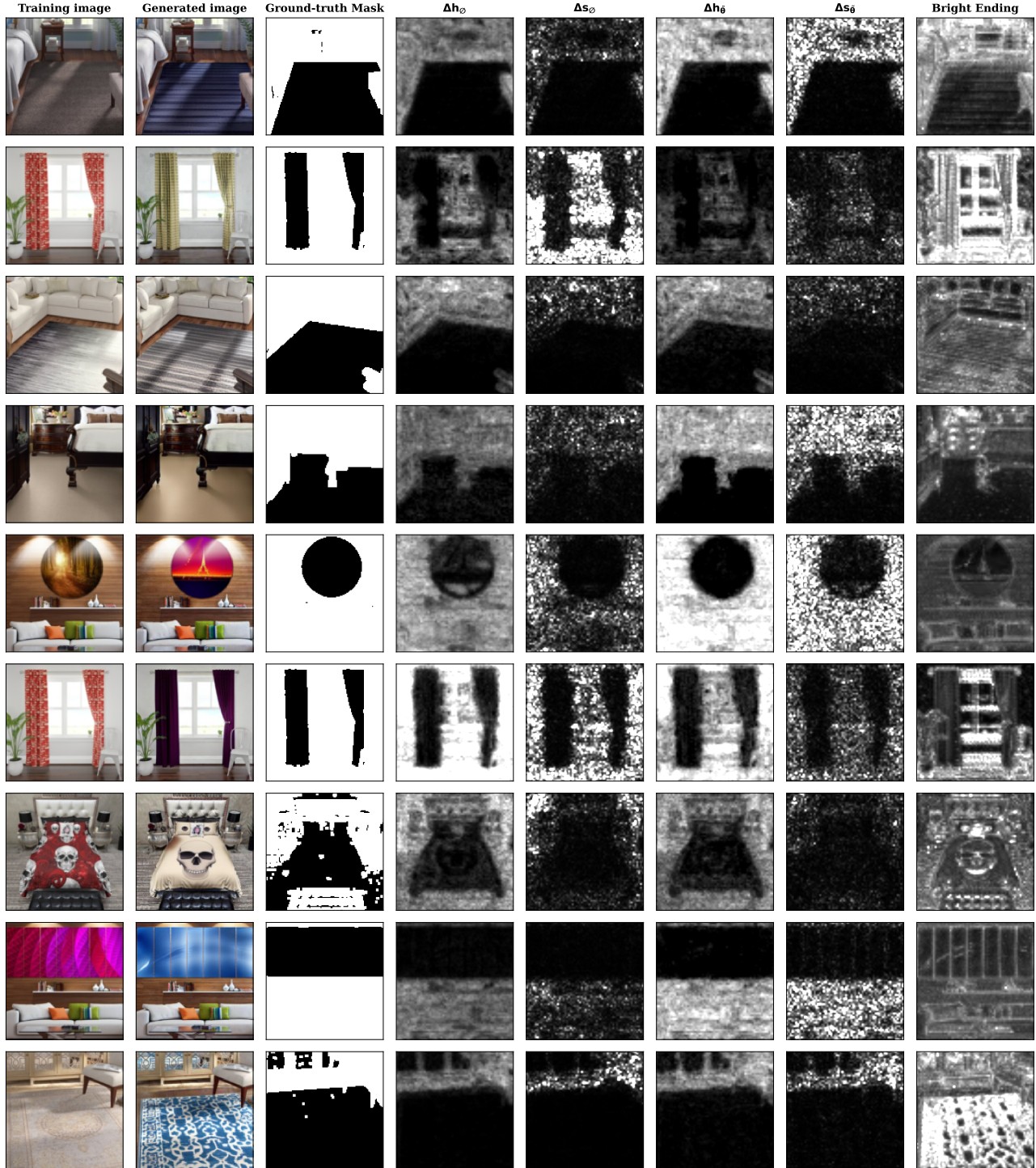

*Figure 4.* **Qualitative results for SD v1.4.**

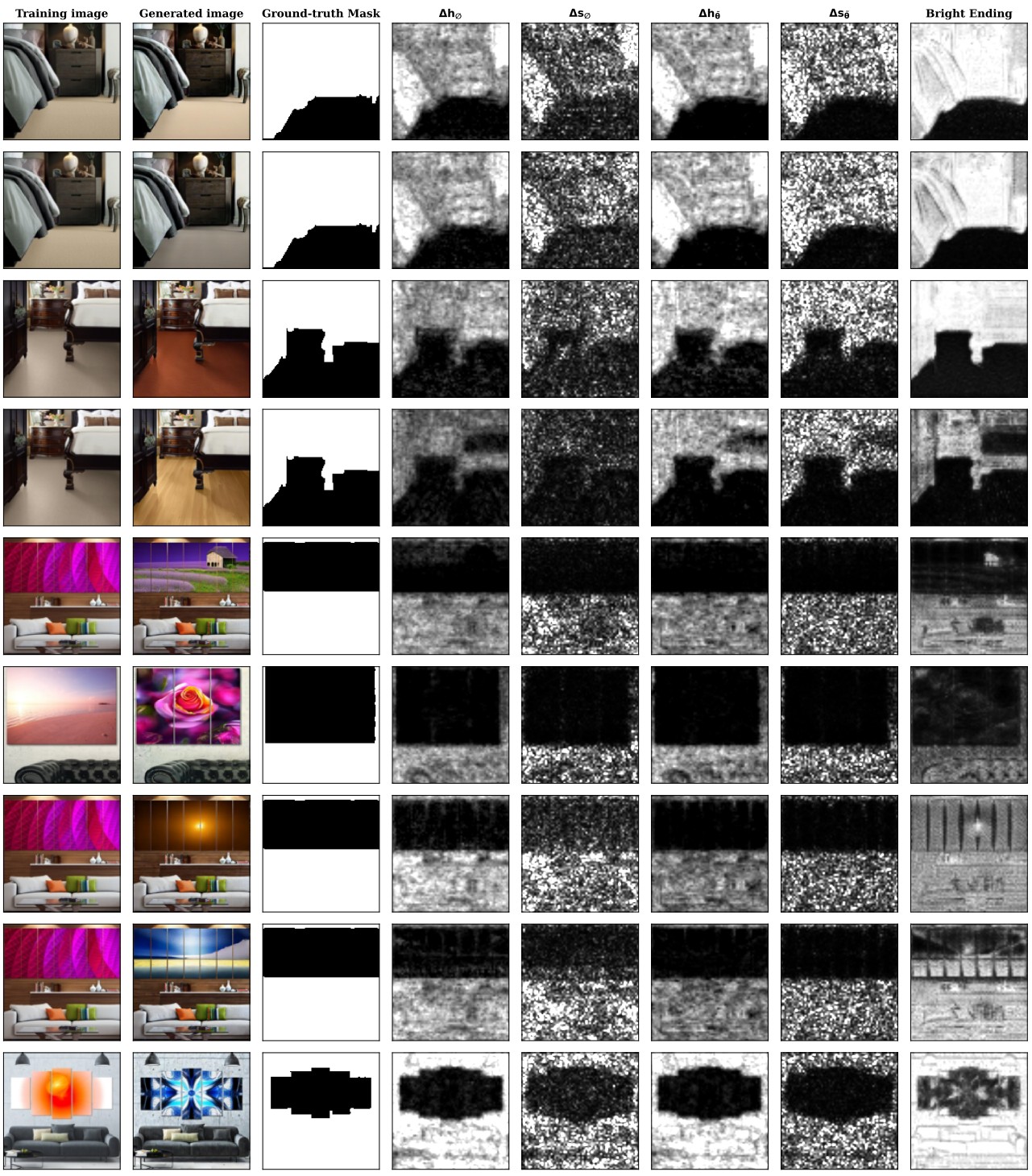

*Figure 5.* **Qualitative results for SD v2.1.**

# B. Extended Related Work - Mitigation

Studies on memorization in diffusion models generally address either detection or mitigation. Given our primary focus on detection, we deferred the discussion of mitigation strategies to this section to provide a comprehensive review of the literature.

Existing mitigation strategies can be broadly categorized based on their point of intervention: conditioning prompt, guidance scale, initial noise, and model parameters. At the **prompt level**, the pioneering work of Somepalli et al. (2023b) proposed replacing or adding random tokens to disrupt the prompt-image association, while Wen et al. (2024) optimized prompt embeddings to minimize their detection metric (Eq. 3). Regarding the **guidance scale**, Jain et al. (2025) identified the "attraction basin" and suggested setting the guidance scale to zero or even applying opposite guidance during early sampling steps to prevent the trajectory from collapsing into memorized states. Concurrent studies on **initial noise** (Jeon et al., 2025; Han et al., 2025) interpret the score-difference-based metric as a measure of sharpness. They propose optimizing the initial noise to minimize this sharpness, thereby preventing semantic loss that often occurs when the prompt itself is modified. Finally, another line of research (Chavhan et al., 2024; Hintersdorf et al., 2024) has found that memorization is typically localized within specific **model parameters or neurons**, particularly in the value layers of cross-attention blocks. To address this, several works (Chavhan et al., 2024; Hintersdorf et al., 2024; Ye et al., 2026; Di et al., 2026) propose pruning or deactivating these localized weights to eliminate memorized content while preserving overall generation quality.

# C. Algorithm

We summarize the full procedure for $\Delta h_\emptyset, \Delta h_{\tilde{\theta}}, \Delta s_\emptyset, \Delta s_{\tilde{\theta}}$ in Algorithm 1. Specifically, it outlines how curvature-difference-based metrics ($\Delta h_\emptyset, \Delta h_{\tilde{\theta}}$) are computed efficiently using the Hutchinson trick, and how score-difference-based metrics ($\Delta s_\emptyset, \Delta s_{\tilde{\theta}}$) further reduce computational overhead.

---

**Algorithm 1** Localizing Memorization via Coordinate-wise Curvature Differences

---

1: **Input:** Prompt $c$, Target step $t^*$, Samples $K$, Models $\theta, \tilde{\theta}$, Baseline $B \in \{\emptyset, \tilde{\theta}\}$, Method $M \in \{\Delta s, \Delta h\}$
2: **Output:** Localization Map $\mathcal{M}$
3: {Sample $x_{t^*}$}
4: $x_T \sim \mathcal{N}(\mathbf{0}, \mathbf{I})$
5: **for** $t = T, T-1, \ldots, t^*+1$ **do**
6: $\quad s_{\text{cfg}} \leftarrow s_\theta(x_t, t, \emptyset) + w \cdot (s_\theta(x_t, t, c) - s_\theta(x_t, t, \emptyset))$ {Sampling with CFG scale $w$}
7: $\quad x_{t-1} \leftarrow \text{SamplerStep}(x_t, t, s_{\text{cfg}})$ {e.g., DDIM}
8: **end for**
9: {Define score difference at $t^*$:}
10: **if** $B = \emptyset$ **then**
11: $\quad s_{\text{diff}} \leftarrow s_\theta(x_{t^*}, t^*, c) - s_\theta(x_{t^*}, t^*, \emptyset)$ {Unconditional baseline}
12: **else if** $B = \tilde{\theta}$ **then**
13: $\quad s_{\text{diff}} \leftarrow s_\theta(x_{t^*}, t^*, c) - s_{\tilde{\theta}}(x_{t^*}, t^*, c)$ {Less-trained baseline}
14: **end if**
15: {Obtain Localization Map}
16: **if** $M = \Delta s$ **then**
17: $\quad \mathbf{V} \leftarrow s_{\text{diff}}^{\odot 2}$ {Score-difference surrogate}
18: **else if** $M = \Delta h$ **then**
19: $\quad \mathbf{d} \leftarrow \mathbf{0}$
20: $\quad$ **for** $k = 1$ **to** $K$ **do**
21: $\quad\quad v_k \sim \mathcal{U}(\{-1, 1\})^{\dim(x_{t^*})}$ {Rademacher random vector}
22: $\quad\quad \mathbf{g} \leftarrow \nabla_{x_{t^*}} (s_{\text{diff}} \cdot v_k)$ {Efficiently computed via VJP trick}
23: $\quad\quad \mathbf{d} \leftarrow \mathbf{d} + v_k \odot \mathbf{g}$ {Hutchinson estimator (Appendix D.1)}
24: $\quad$ **end for**
25: $\quad \mathbf{V} \leftarrow -\mathbf{d}/K$ {Coordinate-wise curvature difference}
26: **end if**
27: $\mathcal{M} \leftarrow \sum_{\text{channel}} \mathbf{V}$ {Spatial map via channel aggregation}
28: **return** $\mathcal{M}$

---

# D. Proofs

## D.1. Hutchinson estimator

**Proposition D.1** (Hutchinson trick for $\mathrm{diag}(A)$). *Let $A \in \mathbb{R}^{n \times n}$. Let $z \in \mathbb{R}^n$ be a random vector whose entries are i.i.d. Rademacher, i.e., $\mathbb{P}(z_i = +1) = \mathbb{P}(z_i = -1) = \frac{1}{2}$, independent across $i$. Define the elementwise (Hadamard) product by $\odot$. Then*

$$\mathbb{E}\big[z \odot (Az)\big] \; = \; \mathrm{diag}(A),$$

*i.e., $z \odot (Az)$ is an unbiased estimator of the diagonal of $A$. Moreover, with i.i.d. samples $\{z^{(k)}\}_{k=1}^{K}$, the Monte Carlo estimator*

$$\widehat{\mathrm{diag}}(A) \; := \; \frac{1}{K} \sum_{k=1}^{K} z^{(k)} \odot \big(A z^{(k)}\big)$$

*satisfies $\mathbb{E}[\widehat{\mathrm{diag}}(A)] = \mathrm{diag}(A)$.*

*Proof.* Fix an index $i \in \{1, \dots, n\}$. The $i$-th component of $z \odot (Az)$ is

$$\big(z \odot (Az)\big)_i = z_i (Az)_i = z_i \sum_{j=1}^{n} A_{ij} z_j = \sum_{j=1}^{n} A_{ij} \, z_i z_j.$$

Taking expectation and using linearity,

$$\mathbb{E}\big[\big(z \odot (Az)\big)_i\big] = \sum_{j=1}^{n} A_{ij} \, \mathbb{E}[z_i z_j].$$

Because $\{z_i\}$ are independent Rademacher variables, we have

$$\mathbb{E}[z_i z_j] = \begin{cases} \mathbb{E}[z_i^2] = 1, & i = j, \\ \mathbb{E}[z_i]\mathbb{E}[z_j] = 0, & i \neq j, \end{cases}$$

since $\mathbb{E}[z_i] = 0$. Therefore only the term $j = i$ remains:

$$\mathbb{E}\big[\big(z \odot (Az)\big)_i\big] = A_{ii}.$$

Since this holds for every $i$, we conclude

$$\mathbb{E}\big[z \odot (Az)\big] = \mathrm{diag}(A).$$

Finally, the sample average $\widehat{\mathrm{diag}}(A)$ is also unbiased by linearity of expectation. $\qquad\square$

**Application to Curvature Differences**   We utilize the linearity of the Hutchinson estimator to efficiently compute the proposed curvature difference metrics. Recall that $\Delta h_\emptyset$ is defined as the difference between the coordinate-wise curvatures of the conditional and unconditional models:

$$\Delta h_\emptyset \; = \; \mathrm{diag}\big(-H_\theta(x_t, c)\big) - \mathrm{diag}\big(-H_\theta(x_t)\big) \; = \; \mathrm{diag}\big(H_\theta(x_t) - H_\theta(x_t, c)\big).$$

By applying Proposition D.1 with $A = H_\theta(x_t) - H_\theta(x_t, c)$, we can estimate this difference using shared random vectors $z$:

$$\hat{\Delta} h_\emptyset \; = \; \frac{1}{K} \sum_{k=1}^{K} z^{(k)} \odot \big((H_\theta(x_t) - H_\theta(x_t, c)) z^{(k)}\big).$$

Using a shared $z$ for both terms allows for a coupled estimation that typically exhibits lower variance than estimating each term independently. This derivation applies analogously to the less-trained baseline metric $\Delta h_{\tilde{\theta}}$, where the unconditional Hessian $H_\theta(x_t)$ is replaced by the less-trained conditional Hessian $H_{\tilde{\theta}}(x_t, c)$.

### D.2. Proof of Proposition 4.1

We provide a formal derivation of the relationship between the Hessian of the log-likelihood and the conditional covariance of the reverse process. Although this result is established in the literature (Kadkhodaie et al., 2024; Ou et al., 2025; Zhang et al., 2023), we include the derivation here for completeness.

**Statement.** Let the forward transition kernel be

$$p(x_t \mid x_0) = \mathcal{N}\left(\alpha_t\, x_0,\, \sigma_t^2 I\right),$$

and define the marginal

$$p(x_t) = \int p(x_t \mid x_0)\, p(x_0)\, dx_0.$$

Then the conditional covariance satisfies

$$\mathrm{Cov}[x_0 \mid x_t] = \frac{1}{\alpha_t^2}\left(\sigma_t^4 \nabla_{x_t}^2 \log p(x_t) + \sigma_t^2 I\right).$$

*Proof.* By Bayes' rule, the posterior distribution is given by $p(x_0 \mid x_t) = p(x_t \mid x_0)p(x_0)/p(x_t)$. The score function of the marginal density $p(x_t)$ can be expressed as:

$$
\begin{aligned}
\nabla_{x_t} \log p(x_t) &= \frac{\nabla_{x_t} p(x_t)}{p(x_t)} \\
&= \frac{1}{p(x_t)} \int \nabla_{x_t} p(x_t \mid x_0)\, p(x_0)\, \mathrm{d}x_0.
\end{aligned}
\tag{7}
$$

Since the forward transition kernel is Gaussian, $p(x_t \mid x_0) = \mathcal{N}(x_t; \alpha_t x_0, \sigma_t^2 I)$, its gradient with respect to $x_t$ is:

$$\nabla_{x_t} p(x_t \mid x_0) = -\frac{x_t - \alpha_t x_0}{\sigma_t^2}\, p(x_t \mid x_0). \tag{8}$$

Substituting (8) into (7), we obtain Tweedie's formula for the posterior mean $\mathbb{E}[x_0 \mid x_t]$:

$$
\begin{aligned}
\nabla_{x_t} \log p(x_t) &= \frac{1}{p(x_t)} \int -\frac{x_t - \alpha_t x_0}{\sigma_t^2}\, p(x_t \mid x_0)p(x_0)\, \mathrm{d}x_0 \\
&= -\frac{x_t}{\sigma_t^2} + \frac{\alpha_t}{\sigma_t^2} \int x_0\, \frac{p(x_t \mid x_0)p(x_0)}{p(x_t)}\, \mathrm{d}x_0 \\
&= -\frac{x_t}{\sigma_t^2} + \frac{\alpha_t}{\sigma_t^2}\, \mathbb{E}[x_0 \mid x_t].
\end{aligned}
\tag{9}
$$

Rearranging this yields the first moment Tweedie's Formula:

$$\mathbb{E}[x_0 \mid x_t] = \frac{1}{\alpha_t}\left(x_t + \sigma_t^2 \nabla_{x_t} \log p(x_t)\right). \tag{10}$$

Next, we examine the Jacobian of the posterior mean. Differentiating (10) with respect to $x_t$:

$$\nabla_{x_t} \mathbb{E}[x_0 \mid x_t] = \frac{1}{\alpha_t}\left(I + \sigma_t^2 \nabla_{x_t}^2 \log p(x_t)\right). \tag{11}$$

Alternatively, we can compute the Jacobian directly using the log-derivative trick. Note that

$$\nabla_{x_t} \log p(x_0 \mid x_t) = \frac{\alpha_t x_0 - x_t}{\sigma_t^2} - \nabla_{x_t} \log p(x_t).$$

Then, taking the gradient of the expectation:

$$\nabla_{x_t}\mathbb{E}[x_0 \mid x_t] = \nabla_{x_t}\int x_0\, p(x_0 \mid x_t)\, \mathrm{d}x_0$$

$$= \int x_0 (\nabla_{x_t} p(x_0 \mid x_t))^\top \, \mathrm{d}x_0$$

$$= \int x_0\, p(x_0 \mid x_t)(\nabla_{x_t}\log p(x_0 \mid x_t))^\top \, \mathrm{d}x_0$$

$$= \int x_0\, p(x_0 \mid x_t)\left(\frac{\alpha_t x_0 - x_t}{\sigma_t^2} - \nabla_{x_t}\log p(x_t)\right)^\top \mathrm{d}x_0. \tag{12}$$

Rearranging Eq. (10), we have $\nabla_{x_t}\log p(x_t) = \frac{\alpha_t\mathbb{E}[x_0|x_t]-x_t}{\sigma_t^2}$. Substituting this relation into the integral simplifies the term inside to $\frac{\alpha_t}{\sigma_t^2}(x_0 - \mathbb{E}[x_0 \mid x_t])^\top$. Therefore:

$$\nabla_{x_t}\mathbb{E}[x_0 \mid x_t] = \frac{\alpha_t}{\sigma_t^2}\int x_0(x_0 - \mathbb{E}[x_0 \mid x_t])^\top p(x_0 \mid x_t)\, \mathrm{d}x_0$$

$$= \frac{\alpha_t}{\sigma_t^2}\int (x_0 - \mathbb{E}[x_0 \mid x_t] + \mathbb{E}[x_0 \mid x_t])(x_0 - \mathbb{E}[x_0 \mid x_t])^\top p(x_0 \mid x_t)\, \mathrm{d}x_0$$

$$= \frac{\alpha_t}{\sigma_t^2}\left(\int (x_0 - \mathbb{E}[x_0 \mid x_t])(x_0 - \mathbb{E}[x_0 \mid x_t])^\top p(x_0 \mid x_t)\, \mathrm{d}x_0\right)$$

$$= \frac{\alpha_t}{\sigma_t^2}\mathrm{Cov}[x_0 \mid x_t]. \tag{13}$$

Finally, by equating (11) and (13), we solve for the covariance:

$$\frac{\alpha_t}{\sigma_t^2}\mathrm{Cov}[x_0 \mid x_t] = \frac{1}{\alpha_t}\left(I + \sigma_t^2\nabla_{x_t}^2\log p(x_t)\right)$$

$$\mathrm{Cov}[x_0 \mid x_t] = \frac{\sigma_t^2}{\alpha_t^2}\left(I + \sigma_t^2\nabla_{x_t}^2\log p(x_t)\right)$$

$$= \frac{1}{\alpha_t^2}\left(\sigma_t^4\nabla_{x_t}^2\log p(x_t) + \sigma_t^2 I\right).$$

$$\square$$

### D.3. Proof of Proposition 4.2

This derivation follows the standard proof of the Fisher information identity (Lehmann & Casella, 1998); we include it here for completeness.

**Statement.** Let $p(c \mid x)$ be twice continuously differentiable with respect to $x$ and assume that differentiation and expectation can be interchanged. Define the Fisher information matrix by

$$I(x) := \mathbb{E}_{c\sim p(c|x)}\left[\nabla_x \log p(c \mid x)\,\nabla_x\log p(c \mid x)^\top\right].$$

Then,

$$I(x) = -\mathbb{E}_{c\sim p(c|x)}\left[\nabla_x^2\log p(c \mid x)\right].$$

*Proof.* Define the score function

$$s(c, x) := \nabla_x\log p(c \mid x).$$

We first note that the expectation of the score vanishes:

$$\mathbb{E}_{c\sim p(c|x)}[s(c,x)] = \int \nabla_x\log p(c \mid x)\,p(c \mid x)\,dc = \int \nabla_x p(c \mid x)\,dc = \nabla_x\int p(c \mid x)\,dc = \nabla_x 1 = 0.$$

Taking the gradient of both sides with respect to $x$ yields

$$0 = \nabla_x \mathbb{E}_{c \sim p(c|x)}[s(c, x)] = \nabla_x \int s(c, x)\, p(c \mid x)\, dc = \int \nabla_x \big(s(c, x)\, p(c \mid x)\big)\, dc,$$

where we used the assumed regularity conditions to interchange differentiation and integration. Applying the product rule, we obtain

$$\nabla_x \big(s\, p\big) = (\nabla_x s)\, p + s\, (\nabla_x p)^\top.$$

Therefore,

$$0 = \int (\nabla_x s(c, x))\, p(c \mid x)\, dc \;+\; \int s(c, x)\, \nabla_x p(c \mid x)^\top\, dc.$$

The first term can be written as

$$\int (\nabla_x s)\, p\, dc = \mathbb{E}_{c \sim p(c|x)}\big[\nabla_x^2 \log p(c \mid x)\big].$$

For the second term, using $\nabla_x p(c \mid x) = p(c \mid x)\, s(c, x)$, we have

$$\int s(c, x)\, (\nabla_x p(c \mid x))^\top\, dc = \int s(c, x)\, s(c, x)^\top\, p(c \mid x)\, dc = \mathbb{E}_{c \sim p(c|x)}\big[s(c, x)\, s(c, x)^\top\big].$$

Combining the two terms, we obtain

$$0 = \mathbb{E}_{c \sim p(c|x)}\big[\nabla_x^2 \log p(c \mid x)\big] + \mathbb{E}_{c \sim p(c|x)}\big[s(c, x)\, s(c, x)^\top\big].$$

Rearranging yields

$$\mathbb{E}_{c \sim p(c|x)}\big[s(c, x)\, s(c, x)^\top\big] = -\mathbb{E}_{c \sim p(c|x)}\big[\nabla_x^2 \log p(c \mid x)\big].$$

By definition of $I(x)$, this completes the proof. □

## E. Analysis of the Underfitted Baseline

In this section, we provide a deeper analysis of the role of the underfitted baseline. Appendix E.1 utilizes a synthetic setup to explain the underlying curvature dynamics that justify subtracting a baseline, while Appendix E.2 empirically investigates how the choice of the baseline checkpoint affects localization performance on Stable Diffusion.

### E.1. Synthetic Experiment on Curvature Dynamics

In Section 5, we observed that our curvature-difference metric effectively localizes memorized regions in SD v2.1, even when using SD v2.0—which is also known to exhibit memorization (Webster, 2023)—as the less-trained baseline $\tilde{\theta}$. In this section, we provide a *partial explanation* for this phenomenon based on the curvature dynamics observed in our synthetic experiment. Note that the setup used in this section is analogous to the Stable Diffusion fine-tuning setup for memorization mitigation experiments in Wen et al. (2024) and Somepalli et al. (2023b).

**Experimental Setup.** To validate our hypothesis that curvature strictly increases with progressive overfitting, we conduct a controlled experiment using a synthetic two-dimensional dataset ($N = 10{,}000$, duplication ratio $\rho = 0.5\%$), as shown in Figure 6. The majority (99.5%) of samples are drawn from a rank-1 noisy manifold:

$$y_{1d} = Az + \eta, \qquad z \sim \mathcal{N}(0, 1), \quad \eta \sim \mathcal{N}(0, \sigma_{\text{data}}^2 I_2), \tag{14}$$

with $A = [0.5, 0]^\top$ and $\sigma_{\text{data}} = 3 \times 10^{-2}$. The remaining 0.5% form a duplicated outlier cluster centered at $x_{\text{dup}} = (2.5, 2.0)$, drawn from an isotropic Gaussian:

$$y_{\text{dup}} \sim \mathcal{N}(x_{\text{dup}}^\star, \sigma_{\text{dup}}^2 I_2), \tag{15}$$

where $\sigma_{\text{dup}} = 10^{-5} \ll \sigma_{\text{data}}$.

We train an unconditional DDPM with an MLP denoiser for 60,000 updates. Let $\sigma_t = \sqrt{1 - \bar{\alpha}_t}$ and $s_\theta(x_t, t) = -\epsilon_\theta(x_t, t)/\sigma_t$. At a fixed evaluation step $t_{\text{eval}} = 3$, we measure curvature only along the first coordinate:

$$\kappa_1(x) := \big(-\nabla_x s_\theta(x, t_{\text{eval}})\big)_{11},$$

evaluated at both the outlier center $x_{\text{dup}}$ and a representative 1d noisy manifold sample $x_{1d}$.

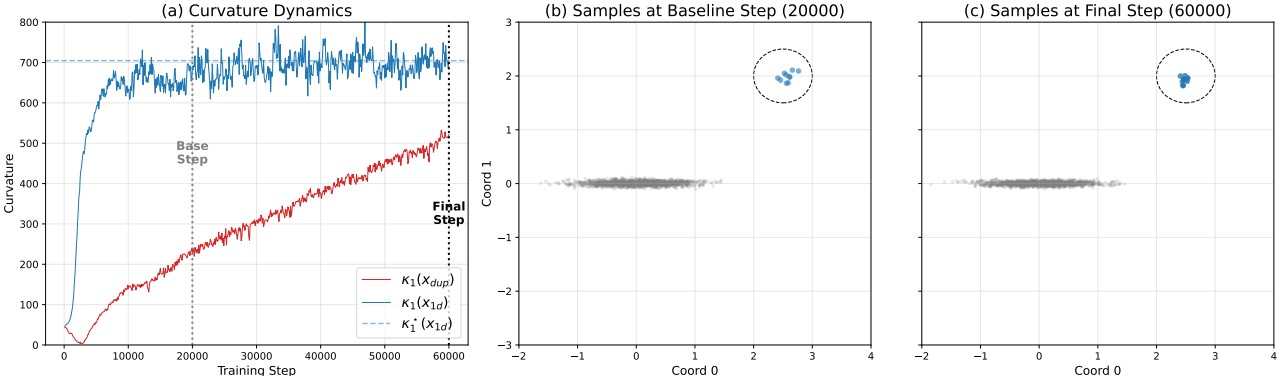

*Figure 6.* Synthetic experiment on curvature dynamics under progressive overfitting. (a) Training trajectories of the coordinate-wise curvature $\kappa_1(x) = (-\nabla_x s_\theta(x, t_{\mathrm{eval}}))_{11}$, measured at the duplicated outlier center $x_{\mathrm{dup}}$ and at a representative rank-1 noisy manifold sample $x_{1\mathrm{d}}$. While $\kappa_1(x_{1\mathrm{d}})$ rapidly saturates at the ground-truth curvature $\kappa_1^\star$, $\kappa_1(x_{\mathrm{dup}})$ continues to increase throughout training. Notably, the model already generates samples from the outlier mode at the baseline checkpoint (20k updates), yet its curvature continues to grow significantly thereafter, indicating progressive sharpening. (b,c) Samples generated by the DDPM at the baseline checkpoint (20k updates) and the final checkpoint (60k updates), respectively. The outlier mode becomes progressively sharper, whereas the rank-1 component remains unchanged.

**Partial Explanation of Curvature Differences.** Based on the results shown in Figure 6, we attribute the effectiveness of our method to the differing curvature dynamics of DD-Mem and OD-Mem. We specifically highlight the interpretation of the curvature trajectory at the outlier mode:

1. **DD-Mem saturates early:** General data manifolds typically possess intrinsic noise ($\sigma_{\mathrm{data}} > 0$) even if the data lies on a low-dimensional manifold (Fefferman et al., 2016). Along the manifold direction, the marginal distribution $p_t(x)$ approximates a convolution of the data distribution $\mathcal{N}(0, \sigma_{\mathrm{data}}^2)$ and the diffusion noise kernel $\mathcal{N}(0, \sigma_t^2)$. Consequently, the curvature—which corresponds to the inverse variance for Gaussians—rapidly converges to the ground-truth curvature $\kappa_1^\star$:

$$\kappa_1^\star \approx (-\nabla^2 \log p_t(x))_{11} \approx \frac{1}{\sigma_{\mathrm{data}}^2 + \sigma_{t_{\mathrm{eval}}}^2}. \tag{16}$$

Since both the less-trained baseline and the fully-trained model reach this saturation point early in training, the curvature difference approaches zero, effectively cancelling out the intrinsic data structure.

2. **OD-Mem continues to sharpen:** As shown in Figure 6, the model successfully generates samples from the outlier mode as early as the baseline checkpoint (20,000 steps). However, crucially, the curvature $\kappa_1$ at this mode does not stop increasing; it continues to rise significantly as training proceeds to the final step (60,000 steps). This indicates that memorization is a process of progressive sharpening. We attribute this primarily to the slow convergence due to data scarcity. Since the outliers constitute only a tiny fraction ($\rho = 0.5\%$) of the training data, the model updates its estimate of the score in this region much less frequently. Furthermore, this prolonged convergence is feasible because the negligible intrinsic variance ($\sigma_{\mathrm{dup}} \ll \sigma_{\mathrm{data}}$) implies a vastly higher theoretical curvature bound, providing ample room for the model to continue sharpening the density well beyond the baseline checkpoint. Therefore, the fully trained model $\theta$ exhibits a significantly sharper landscape than the baseline $\tilde{\theta}$, yielding a large positive curvature difference.

**Justification of using late sampling steps** To justify the use of the final sampling step, we analyze how curvature dynamics vary across timesteps ($t = 20, 200, 800$) in Figure 7. We observe that the data scarcity of the outlier ($\rho = 0.5\%$) consistently leads to slower convergence compared to the 1d manifold, regardless of the timestep. However, at intermediate and high noise levels, the diffusion noise imposes a strictly low theoretical curvature bound ($\kappa \approx 1/\sigma_t^2$). Consequently, despite the slower learning rate, the memorized model quickly hits this low ceiling and saturates. This premature saturation masks the meaningful curvature growth between the baseline and final training steps. Crucially, at late sampling steps (e.g., $t = 20$) where the noise bound is lifted, the model can continue to sharpen the density around the scarce outlier, rendering the overfitting-driven curvature divergence significantly more pronounced.

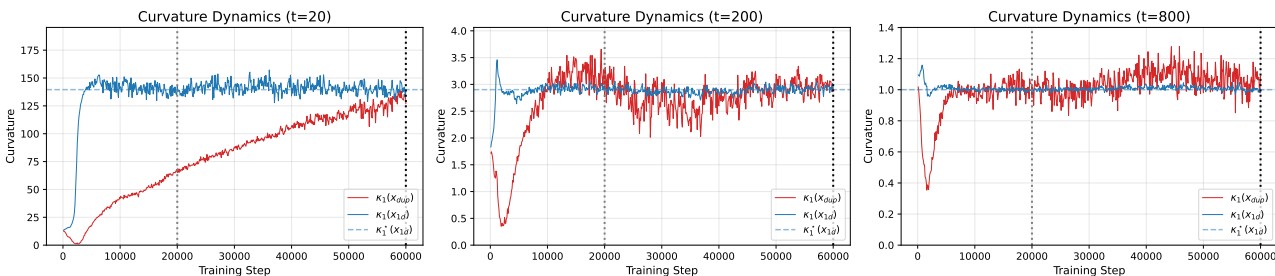

*Figure 7.* Synthetic experiment on Curvature dynamics of $\kappa_1$ at timesteps $t = 20, 200, 800$.

### E.2. Sensitivity to the Choice of the Baseline Model

To evaluate the robustness of our framework with respect to the choice of the less-trained baseline ($\tilde{\theta}$), we conduct an additional ablation study using Stable Diffusion v1.2 as an alternative baseline for the SD v1.4 target model. While SD v1.1 (used in our main experiments) represents an early pre-training stage, SD v1.2 has undergone further fine-tuning, making it a more trained model whose probability landscape is closer to that of the target.

*Table 3.* Baseline checkpoint sensitivity for SD v1.4. We report the performance of difference-based metrics using a more trained model (SD v1.2) as the baseline.

| Method | SD v1.4 (Baseline: SD v1.2) | |
| --- | --- | --- |
| | **TV only (IoU / ACC)** | **All (IoU / ACC)** |
| $\Delta h_{\tilde{\theta}}$ | 0.875 / 0.918 | 0.854 / 0.955 |
| $\Delta s_{\tilde{\theta}}$ | 0.886 / 0.926 | 0.837 / 0.955 |

As shown in Table 3, utilizing SD v1.2 as the baseline yields localization performance that remains highly competitive. As discussed in the remark in Section 5.1 and Appendix E.1, this demonstrates that the baseline is not required to be strictly underfitted; it only needs to be relatively less overfitted than the target model for the curvature-difference metric to successfully capture incremental, overfitting-driven curvature.

However, while our access to intermediate checkpoints is limited, we conjecture that extreme baseline choices may weaken the efficacy of our difference-based methods. For instance, excessively early checkpoints might fail to capture the general data distribution, whereas checkpoints too close to the target may already capture some overfitting-driven curvature, thereby reducing the distinctiveness of the memorization signal.

## F. Additional Details on Experiments in Section 5

### F.1. Details of Less-Trained Baseline Models

In our experiments, we utilize Stable Diffusion (SD) v1.1 and SD v2.0 as less-trained baselines ($\tilde{\theta}$) for the target models SD v1.4 and SD v2.1, respectively. These baselines allow us to isolate the curvature induced by overfitting, as they share the same architecture and lineage but differ significantly in training duration and data exposure.

**SD v1** Stable Diffusion v1.4[3] is a direct continuation of the v1.1 training lineage, with a clearer emphasis on aesthetic quality. Stable Diffusion v1.1 (Baseline) corresponds to the initial pre-training stage and was trained for approximately **431,000 steps** on the LAION-2B dataset. Its primary objective was resolution scaling, progressing from $256 \times 256$ to $512 \times 512$, with minimal aesthetic-based filtering applied to the training data. In contrast, Stable Diffusion v1.4 (Target) resumes from the v1 series checkpoints and undergoes extensive fine-tuning through the v1.2 and v1.4 phases, accumulating approximately **740,000 additional steps**. This fine-tuning is performed on aesthetically filtered subsets of the original dataset.

---

[3] https://huggingface.co/CompVis/stable-diffusion-v1-4

**SD v2**   In the Stable Diffusion v2[4] series, the key distinction between versions arises from differences in safety filtering and the resulting amount of additional training. Stable Diffusion v2.0 (Baseline) was trained under a strict safety constraint ($p_{unsafe} < 0.1$), which ensured safer outputs. Stable Diffusion v2.1 (Target), on the other hand, fine-tunes the v2.0 checkpoint for an additional **210,000 steps** on the same dataset with a substantially relaxed safety filter ($p_{unsafe} < 0.98$).

## F.2. Details on ground-truth mask evaluation in Section 5.1

**Datasets.**   The evaluation unit is a single generated image instance defined by a specific prompt-seed pair, where a single prompt may be associated with up to four seeds matching distinct templates. We assess performance in two scenarios: (1) **TV Only**, which uses all Template Verbatim instances (355 for SD v1.4, 590 for SD v2.1); and (2) **Combined (TV + MV + Nmem)**, which includes Memorized Visual and Non-memorized instances to evaluate discriminative power. In the Combined setting, we enforce class balancing by random subsampling to the minimum category size: for SD v1.4, classes are balanced to 231 instances each (total 693), and for SD v2.1, to 398 instances each (total 796) with no global memorization case. For non-memorized prompts, we used the same dataset as in the detection experiment in Section 5.2.

**Evaluation.**   We resize all metric maps to $256 \times 256$ using bilinear interpolation to match the resolution of the ground-truth masks. We perform a global normalization to $[0, 1]$ independently for each metric over all evaluation samples. We computed the best Intersection-over-Union (IoU) and Pixel Accuracy (ACC) by sweeping a uniform decision threshold $\tau$ over the $[0, 1]$ range with a step size of 0.001 (1001 steps).

## F.3. Implementation Details on Bright Ending (Chen et al., 2025)

We directly adopt the official codebase of Chen et al. (2025) to ensure exact reproduction of their method. We extract the cross-attention maps from the down-sampling blocks of the U-Net at the final denoising step ($t = 1$). Typically, the U-Net contains attention operations at multiple resolutions; we specifically select the maps with a spatial resolution of $64 \times 64$, matching the latent feature size. These maps are first averaged across all attention heads within each layer and then averaged across the selected layers to obtain a single spatial map. From this aggregated map, we extract the channel corresponding to the End-of-Sequence (EOS) token, as it is known to aggregate global semantic information. The resulting $64 \times 64$ map is upsampled to $256 \times 256$ to match the ground-truth mask.

---

[4] https://huggingface.co/Manojb/stable-diffusion-2-1-base

## G. Additional Experiment on Realistic Vision v5.1

To further validate the generalizability of our proposed framework, we extend our evaluation to **Realistic Vision v5.1** [5]. We employ the same evaluation protocol as described in Section 5, utilizing the ground-truth memorization masks.

*Table 4.* Ground-truth mask evaluation on Realistic Vision v5.1

| | Realistic Vision v5.1 | | | |
| | TV only | | All | |
| Method | IoU | ACC | IoU | ACC |
|---|---|---|---|---|
| All-ones | 0.561 | 0.561 | 0.519 | 0.519 |
| Chen et al. (2025) | 0.777 | 0.818 | 0.578 | 0.857 |
| $\Delta h_\emptyset$ | 0.933 | 0.961 | **0.959** | **0.978** |
| $\Delta s_\emptyset$ | 0.879 | 0.929 | 0.906 | 0.953 |
| $\Delta h_{\tilde\theta}$ | **0.939** | **0.964** | 0.895 | 0.971 |
| $\Delta s_{\tilde\theta}$ | 0.864 | 0.917 | 0.619 | 0.874 |

The quantitative results are summarized in the table above. Consistent with the observations in Section 5, our curvature-difference and score-difference metrics generally outperform the prior attention-based method (Chen et al., 2025), in terms of both IoU and Pixel Accuracy.

## H. Ablation Studies for timesteps and Hutchinson Samples

Our ablation studies (Tables 6–9) show that localization performance consistently improves during the later stages of the sampling process. This is consistent with our analysis in Appendix E.1, showing that the curvature gap effectively vanishes under high diffusion noise. Regarding the number of Hutchinson samples $K$, while larger values yield slight improvements, we observe that even a single sample ($K = 1$) achieves competitive accuracy with diminishing returns.

In Table 5, we also report the time overhead for estimating the curvature-based metrics ($\Delta h_\emptyset$, $\Delta h_{\tilde\theta}$) for a single generated image, as a function of the number of Hutchinson samples $K$. All measurements were conducted on a single NVIDIA L40S GPU (48GB). The increased computational cost for $\Delta h_{\tilde\theta}$ compared to $\Delta h_\emptyset$ is due to the backpropagation requiring traversal through two distinct U-Net models ($s_\theta$ and $s_{\tilde\theta}$). In contrast, $\Delta h_\emptyset$ involves backpropagation through only a single U-Net instance, allowing for more efficient computation.

*Table 5.* Computational Overhead by Number of Hutchinson Samples ($K$)

| Model | Method | Sampling Time (50 steps) | Additional Time (Seconds) | | | | |
|---|---|---|---|---|---|---|---|
| | | | K=1 | K=2 | K=4 | K=8 | K=16 |
| SD v1.4 | $\Delta h_\emptyset$ | 1.08 | +0.03 | +0.05 | +0.11 | +0.21 | +0.42 |
| | $\Delta h_{\tilde\theta}$ | | +0.05 | +0.10 | +0.21 | +0.41 | +0.82 |
| SD v2.1 | $\Delta h_\emptyset$ | 1.04 | +0.03 | +0.05 | +0.09 | +0.19 | +0.38 |
| | $\Delta h_{\tilde\theta}$ | | +0.05 | +0.09 | +0.18 | +0.36 | +0.73 |

---

[5] https://civitai.com/

*Table 6.* Ablation Studies for $\Delta h_\emptyset^t$

| | | SD v1.4 | | | | SD v2.1 | | | |
| | | TV only | | All | | TV only | | TV + Non-mem | |
| Timestep | K | IoU | ACC | IoU | ACC | IoU | ACC | IoU | ACC |
|---|---|---|---|---|---|---|---|---|---|
| - | All-ones | 0.560 | 0.560 | 0.522 | 0.522 | 0.649 | 0.649 | 0.325 | 0.325 |
| - | All-zeros | 0.000 | 0.440 | 0.333 | 0.478 | 0.000 | 0.351 | 0.500 | 0.675 |
| 10 | 1 | **0.560** | **0.593** | **0.522** | 0.705 | **0.649** | **0.649** | **0.500** | 0.699 |
| | 2 | **0.560** | 0.592 | **0.522** | 0.720 | **0.649** | **0.649** | **0.500** | 0.707 |
| | 4 | **0.560** | 0.586 | **0.522** | 0.733 | **0.649** | **0.649** | **0.500** | 0.713 |
| | 8 | **0.560** | 0.582 | **0.522** | 0.748 | **0.649** | **0.649** | **0.500** | 0.716 |
| | 16 | **0.560** | 0.580 | **0.522** | **0.764** | **0.649** | **0.649** | **0.500** | **0.718** |
| 20 | 1 | **0.560** | 0.639 | **0.522** | 0.784 | **0.649** | **0.649** | **0.500** | 0.700 |
| | 2 | **0.560** | 0.646 | **0.522** | 0.800 | **0.649** | **0.649** | **0.500** | 0.709 |
| | 4 | **0.560** | 0.650 | **0.522** | 0.814 | **0.649** | **0.649** | **0.500** | 0.716 |
| | 8 | **0.560** | 0.659 | **0.522** | 0.828 | **0.649** | **0.649** | **0.500** | 0.725 |
| | 16 | **0.560** | **0.666** | **0.522** | 0.840 | **0.649** | **0.649** | **0.500** | 0.729 |
| 30 | 1 | 0.560 | 0.658 | 0.522 | 0.844 | 0.649 | 0.669 | **0.500** | 0.739 |
| | 2 | 0.560 | **0.661** | 0.522 | 0.855 | 0.649 | 0.680 | **0.500** | 0.758 |
| | 4 | 0.560 | 0.661 | 0.522 | 0.864 | 0.649 | 0.698 | **0.500** | 0.779 |
| | 8 | 0.560 | 0.658 | 0.523 | 0.869 | 0.649 | 0.715 | **0.500** | 0.801 |
| | 16 | **0.561** | 0.654 | **0.540** | **0.871** | **0.653** | **0.732** | **0.500** | **0.819** |
| 40 | 1 | 0.648 | 0.744 | 0.558 | 0.902 | 0.684 | 0.757 | **0.500** | 0.831 |
| | 2 | 0.680 | 0.765 | 0.609 | 0.913 | 0.725 | 0.793 | **0.500** | 0.857 |
| | 4 | 0.710 | 0.783 | 0.640 | 0.922 | 0.768 | 0.829 | **0.500** | 0.885 |
| | 8 | 0.733 | 0.798 | 0.672 | 0.929 | 0.800 | 0.855 | **0.500** | 0.906 |
| | 16 | **0.747** | **0.808** | **0.700** | **0.933** | **0.823** | **0.873** | **0.500** | **0.920** |
| 49 | 1 | 0.884 | 0.930 | 0.943 | 0.964 | 0.894 | 0.928 | 0.806 | 0.961 |
| | 2 | 0.905 | 0.943 | 0.955 | 0.970 | 0.922 | 0.949 | 0.835 | 0.972 |
| | 4 | 0.913 | 0.948 | 0.959 | 0.972 | 0.934 | 0.958 | 0.853 | 0.977 |
| | 8 | 0.918 | 0.951 | 0.962 | 0.973 | 0.941 | 0.962 | 0.871 | 0.980 |
| | 16 | **0.920** | **0.952** | **0.963** | **0.974** | **0.943** | **0.964** | **0.879** | **0.981** |

*Table 7.* Ablation Studies for $\Delta s_\emptyset^t$

| | SD v1.4 | | | | SD v2.1 | | | |
| | TV only | | All | | TV only | | TV + Non-mem | |
| Timestep | IoU | ACC | IoU | ACC | IoU | ACC | IoU | ACC |
|---|---|---|---|---|---|---|---|---|
| All-ones | 0.560 | 0.560 | 0.522 | 0.522 | 0.649 | 0.649 | 0.325 | 0.325 |
| All-zeros | 0.000 | 0.440 | 0.333 | 0.478 | 0.000 | 0.351 | 0.500 | 0.675 |
| 0 | 0.560 | 0.685 | 0.568 | 0.673 | 0.649 | 0.664 | 0.697 | 0.830 |
| 10 | 0.560 | 0.704 | 0.522 | 0.761 | 0.678 | 0.783 | 0.500 | 0.820 |
| 20 | 0.738 | 0.834 | 0.522 | 0.816 | 0.649 | 0.757 | 0.500 | 0.837 |
| 30 | 0.716 | 0.821 | 0.562 | 0.917 | 0.649 | 0.649 | 0.500 | 0.790 |
| 40 | 0.814 | 0.874 | 0.643 | 0.940 | **0.890** | **0.920** | 0.500 | **0.932** |
| 49 | **0.837** | **0.902** | **0.927** | **0.953** | 0.784 | 0.840 | **0.806** | 0.920 |

*Table 8.* Ablation Studies for $\Delta h_{\tilde{\theta}}^{t}$

| | | SD v1.4 | | | | SD v2.1 | | | |
| | | TV only | | All | | TV only | | TV + Non-mem | |
| Timestep | K | IoU | ACC | IoU | ACC | IoU | ACC | IoU | ACC |
|---|---|---|---|---|---|---|---|---|---|
| - | All-ones | 0.560 | 0.560 | 0.522 | 0.522 | 0.649 | 0.649 | 0.325 | 0.325 |
| - | All-zeros | 0.000 | 0.440 | 0.333 | 0.478 | 0.000 | 0.351 | 0.500 | 0.675 |
| 10 | 1 | 0.572 | 0.673 | 0.522 | 0.792 | **0.649** | 0.649 | **0.500** | 0.717 |
| | 2 | 0.592 | 0.698 | 0.522 | 0.813 | **0.649** | 0.654 | **0.500** | 0.729 |
| | 4 | 0.622 | 0.726 | 0.522 | 0.838 | **0.649** | 0.656 | **0.500** | 0.740 |
| | 8 | 0.659 | 0.758 | 0.535 | 0.856 | **0.649** | 0.658 | **0.500** | 0.756 |
| | 16 | **0.688** | **0.785** | **0.586** | **0.866** | **0.649** | **0.679** | **0.500** | **0.772** |
| 20 | 1 | 0.702 | 0.804 | 0.531 | 0.871 | **0.649** | **0.649** | **0.500** | 0.689 |
| | 2 | 0.752 | 0.839 | 0.550 | 0.898 | **0.649** | **0.649** | **0.500** | 0.701 |
| | 4 | 0.795 | 0.869 | 0.566 | 0.917 | **0.649** | **0.649** | **0.500** | 0.710 |
| | 8 | 0.825 | 0.889 | 0.586 | 0.929 | **0.649** | **0.649** | **0.500** | 0.726 |
| | 16 | **0.843** | **0.901** | **0.638** | **0.936** | **0.649** | **0.649** | **0.500** | **0.735** |
| 30 | 1 | 0.794 | 0.865 | 0.579 | 0.912 | 0.671 | 0.724 | **0.500** | 0.723 |
| | 2 | 0.831 | 0.891 | 0.593 | 0.933 | 0.698 | 0.753 | **0.500** | 0.743 |
| | 4 | 0.856 | 0.908 | 0.602 | 0.945 | 0.720 | 0.782 | **0.500** | 0.772 |
| | 8 | 0.871 | 0.918 | 0.618 | 0.952 | 0.741 | 0.806 | **0.500** | 0.805 |
| | 16 | **0.879** | **0.923** | **0.665** | **0.955** | **0.765** | **0.824** | **0.500** | **0.827** |
| 40 | 1 | 0.863 | 0.914 | 0.607 | 0.943 | 0.757 | 0.816 | **0.500** | 0.812 |
| | 2 | 0.889 | 0.932 | 0.617 | 0.956 | 0.802 | 0.857 | **0.500** | 0.850 |
| | 4 | 0.904 | 0.941 | 0.721 | 0.963 | 0.836 | 0.882 | **0.500** | 0.882 |
| | 8 | 0.911 | 0.945 | 0.772 | 0.967 | 0.855 | 0.898 | **0.500** | 0.906 |
| | 16 | **0.914** | **0.947** | **0.790** | **0.969** | **0.867** | **0.907** | **0.500** | **0.920** |
| 49 | 1 | 0.919 | 0.951 | 0.791 | 0.962 | 0.912 | 0.942 | 0.707 | 0.968 |
| | 2 | 0.929 | 0.957 | 0.844 | 0.967 | 0.929 | 0.955 | 0.771 | 0.975 |
| | 4 | 0.933 | 0.959 | 0.876 | 0.969 | 0.939 | 0.961 | 0.808 | 0.979 |
| | 8 | 0.936 | 0.961 | **0.891** | **0.971** | 0.945 | 0.965 | 0.836 | 0.982 |
| | 16 | **0.936** | **0.961** | 0.887 | 0.956 | **0.947** | **0.967** | **0.852** | **0.983** |

*Table 9.* Ablation Studies for $\Delta s_{\tilde{\theta}}^{t}$

| | SD v1.4 | | | | SD v2.1 | | | |
| | TV only | | All | | TV only | | TV + Non-mem | |
| Timestep | IoU | ACC | IoU | ACC | IoU | ACC | IoU | ACC |
|---|---|---|---|---|---|---|---|---|
| All-ones | 0.560 | 0.560 | 0.522 | 0.522 | 0.649 | 0.649 | 0.325 | 0.325 |
| All-zeros | 0.000 | 0.440 | 0.333 | 0.478 | 0.000 | 0.351 | 0.500 | 0.675 |
| 0 | 0.560 | 0.710 | 0.583 | 0.682 | 0.670 | 0.779 | 0.795 | 0.888 |
| 10 | 0.836 | 0.893 | **0.710** | 0.907 | 0.688 | 0.792 | 0.500 | 0.897 |
| 20 | 0.860 | 0.909 | 0.608 | **0.934** | 0.649 | 0.736 | 0.500 | 0.854 |
| 30 | 0.876 | 0.920 | 0.601 | 0.934 | 0.807 | 0.867 | 0.500 | 0.904 |
| 40 | **0.883** | **0.927** | 0.597 | 0.929 | 0.889 | 0.922 | 0.500 | 0.936 |
| 49 | 0.874 | 0.924 | 0.651 | 0.909 | **0.920** | **0.947** | **0.880** | **0.974** |

