# OpenReview forum: "Localizing Memorized Regions in Diffusion Models via Coordinate-Wise Curvature Differences"
_ICML.cc/2026/Conference — ICML 2026 regular_

### Official Review · Reviewer_bHtM · 2026-03-09

**Soundness:** 2
**Presentation:** 3
**Significance:** 3
**Originality:** 3
**Overall Recommendation:** 4
**Confidence:** 4

**Summary:**

This paper studies memorization detection in diffusion models. Compared to previous methods, the authors want to achieve a method that is local, model-agnostic, and generally better. To this end, this paper first links memorization to low local variance, then links low local variance to high curvature, and further isolates undesirable memorization from data-distribution driven one. The authors also propose to use score-difference as an efficient surrogate. The proposed method is validated on two models experimentally.

**Compliance With Llm Reviewing Policy:**

Affirmed.

**Final Justification:**

I apologize for my previous misunderstanding. In this case, I acknowledge the practicality of this study and will raise my scores accordingly.

**Key Questions For Authors:**

1. Have the authors actually experimented using unconditional generation to isolate data-distribution driven memorization? Compared to using a less-trained version, unconditional generation would be much more practical.

**Limitations:**

yes

**Strengths And Weaknesses:**

Pros:
1. The overall presentation is clear and easy to follow.
2. The proposed method is reasonable, well motivated, and supported by relatively strong theoretical base.

Cons:
1. (Major) To isolate undesirable memorization from data-distribution-driven one, the authors use a less-trained version of the same model. However, this is highly impractical in real-world, as most large-scale t2i models do not release less trained versions. (Since the authors mainly experiment with SD v1.4 and SD v2.1, I suppose this method is intended to be applied to real-world models.)
2. (Minor) The niche of this work feels a bit narrow. Specifically, to understand why the method is novel, one needs to first understand two lines of previous work. Nevertheless, I can still sense the novelty of this study, though it takes more effort.
3. (Minor) Presentation-wise, when the authors propose to use the diagonal of the Hessian in the last paragraph of section 4.2, the narrative feels a bit too brief. This part could use a more detailed explanation of why the diagonal of the Hessian can serve this purpose.

---

> ### Author Rebuttal · Authors · 2026-03-30
>
> We sincerely appreciate your valuable feedback.
>
> Below, we address your concerns, in particular clarifying **a key misunderstanding regarding the major concern**.
> >(Major) To isolate undesirable memorization from data-distribution-driven one, the authors use a less-trained version of the same model. However, this is highly impractical in real-world, as most large-scale t2i models do not release less trained versions.
>
> > **Have the authors actually experimented using unconditional generation to isolate data-distribution driven memorization?** Compared to using a less-trained version, unconditional generation would be much more practical.
>
> We would like to clarify that our method does **not** rely on a less-trained model, as **all main results are already obtained using the unconditional baseline**, while the less-trained model is included as an **optional** alternative to further support our framework.
>
> Specifically, **all figures and tables** report results based on **the unconditional baseline ($\Delta h_{\emptyset}$ (Eq. 1) and $\Delta s_{\emptyset}$ (Eq. 4))**, which is readily available in standard diffusion pipelines and more practical, as you pointed out. The methods using a less-trained baseline ($\Delta h_{\tilde{\theta}}$ and $\Delta s_{\tilde{\theta}}$) are included for additional validation.
>
> Therefore, our method is directly applicable to real-world large-scale diffusion models, even when intermediate checkpoints are not available.
>
> In Section 4.3, we introduce the unconditional baseline first, followed by the less-trained baseline. This may have led to some confusion for the reviewer. We will revise the paper to make this distinction more explicit and avoid any ambiguity.
>
>
> >(Minor) The niche of this work feels a bit narrow. Specifically, to understand why the method is novel, one needs to first understand two lines of previous work. Nevertheless, I can still sense the novelty of this study, though it takes more effort.
>
> We thank the reviewer for acknowledging the novelty of our work. While our contributions are already outlined in the Introduction and Related Work sections, we briefly summarize the key points here to improve clarity and will further clarify them in the revision.
>
> - Extension of geometric frameworks ([1,2]). Prior works such as [1,2] characterize memorization using global geometric quantities (e.g., LID or sharpness), which capture how many degrees of freedom collapse. In contrast, we extend this line of work by introducing a coordinate-wise perspective, which reveals where memorization occurs within a sample.
>
> - Novel interpretation of score-difference metrics ([4]). While [4] proposes a practical detection metric, its underlying mechanism remains largely heuristic. Prior works [1,2] also attempt to interpret this metric from a geometric perspective; however, they do not clearly explain why the gap is meaningful or why the unconditional model is an appropriate baseline. We provide a novel interpretation clarifying the roles of the gap and the unconditional baseline.
>
> - Improved localization over attention-based methods ([3]). Compared to [3], which relies on model-specific attention signals, our approach is model-agnostic and grounded in geometry, and achieves more accurate and consistent localization of memorized regions.
>
> [1] Ross et al. "A Geometric Framework for Understanding Memorization in Generative Models", ICLR, 2025.
>
> [2] Jeon et al. "Understanding and Mitigating Memorization in Generative Models via Sharpness of Probability Landscapes", ICML, 2025.
>
> [3] Chen et al. "Exploring Local Memorization in Diffusion Models via Bright Ending Attention", ICLR, 2025.
>
> [4] Wen et al. "Detecting, Explaining, and Mitigating Memorization in Diffusion Models", ICLR, 2024.
>
> >(Minor) Presentation-wise, when the authors propose to use the diagonal of the Hessian in the last paragraph of section 4.2, the narrative feels a bit too brief. This part could use a more detailed explanation of why the diagonal of the Hessian can serve this purpose.
>
> As discussed in Section 4.1, local memorization manifests as low variance along specific coordinate directions. Proposition 4.1 links this variance to curvature, showing that low-variance directions correspond to high curvature.
>
> Therefore, the diagonal of the Hessian, which captures coordinate-wise curvature, provides a direct way to detect such variance collapse at each coordinate, and thus naturally enables spatial localization of memorized regions.
>
> We will revise Section 4.2 to include a more detailed explanation of this connection, to improve clarity for the reader.

---

> > ### Author Rebuttal · Reviewer_bHtM · 2026-04-03
> >
> > I apologize for my previous misunderstanding. In this case, I acknowledge the practicality of this study and will raise my scores accordingly.
> >
> > As for my W2, actually I wanted to say that the presentation of the introduction section might use some refinement to make it easier for readers to quickly grasp the niche of this study.

---

> > > ### Author Response · Authors · 2026-04-03
> > >
> > > We sincerely thank you for your thoughtful reconsideration and positive assessment of our work. We sincerely appreciate your feedback and will refine the introduction in the final version to improve clarity.

---

### Official Review · Reviewer_nvQ6 · 2026-03-11

**Soundness:** 3
**Presentation:** 3
**Significance:** 3
**Originality:** 4
**Overall Recommendation:** 5
**Confidence:** 3

**Summary:**

The authors formulate local memorization as variance collapse along coordinate dimensions and further relate it to the coordinate-wise curvature of the log-density. Based on this formulation, they propose curvature-difference and score-difference indicators with respect to an underfitted baseline. By subtracting the curvature of an unconditional model or a less-trained checkpoint, the proposed method aims to remove high-curvature regions caused by the intrinsically low-variance structure of the data, thereby focusing more specifically on memorization regions driven by overfitting.

**Compliance With Llm Reviewing Policy:**

Affirmed.

**Final Justification:**

Thank you to the authors for their thoughtful revisions and clarifications, which have improved the overall quality and presentation of the paper. The work addresses an interesting problem and demonstrates solid technical effort. However, I still have some remaining concerns, and therefore maintain my original score.

**Key Questions For Authors:**

1. It would be preferable to include additional experiments analyzing the sensitivity of the proposed method to the guidance scale and the baseline checkpoint.
2. For concept-level memorization, have the authors observed that the method fails systematically, or does it still produce some weaker but interpretable response pattern?
3. Please include more baseline methods.
4. Please add a detailed execution pipeline diagram for the proposed method.

**Limitations:**

Yes

**Strengths And Weaknesses:**

Strength:
1. The paper addresses a clear and important problem, namely memorization detection in diffusion models.
2. The experimental evaluation is well targeted, using ground-truth masks for spatial localization assessment rather than relying solely on external detection performance.

Weakness:
1. The paper lacks sensitivity analysis, for example regarding the sensitivity of the proposed method to the guidance scale and the baseline checkpoint.
2. Comparisons with existing methods remain somewhat limited. For local localization, the paper mainly compares against Bright Ending and raw curvature; a richer set of alternative baselines or ablation studies could be added.
3. The paper lacks a rigorous execution pipeline diagram, which reduces its reproducibility.

---

> ### Author Rebuttal · Authors · 2026-03-30
>
> We sincerely appreciate your valuable feedback. Below, we address specific points you raised.
>
> > Comparisons with existing methods remain somewhat limited. For local localization, the paper mainly compares against Bright Ending and raw curvature; a richer set of alternative baselines or ablation studies could be added.
>
> > Please include more baseline methods.
>
> Thank you for the helpful suggestion. We agree that including additional baselines can further strengthen the empirical evaluation.
>
> To clarify, our work focuses on *spatial localization* of memorized regions. To the best of our knowledge, Bright Ending [1] is the only prior work that explicitly addresses this problem.
>
> Following the suggestion, we experimented with a simple score-based baseline using the per-pixel squared magnitude of the conditional score, $s_\theta(x_t,c)^{\odot 2}$, to compare with $\Delta s$.
>
> | Metric (IoU / ACC) | TV only | All |
> |--------------------|---------------|---------------|
> | SD v1.4            | 0.762 / 0.846 | 0.536 / 0.790 |
> | SD v2.1            | 0.785 / 0.845 | 0.500 / 0.837 |
>
> We observe that this simple baseline performs worse than $\Delta s$ and does not change our main conclusions. We will include this comparison in the final version.
>
> [1] Chen et al. "Exploring Local Memorization in Diffusion Models via Bright Ending Attention", ICLR, 2025.
>
> > It would be preferable to include additional experiments analyzing the sensitivity of the proposed method to the guidance scale and the baseline checkpoint.
>
> We thank the reviewer for the helpful suggestion.
>
> **(1) Guidance scale sensitivity**
>
> We evaluate our method under a lower guidance scale (2.0) in addition to the default setting reported in the paper. The results are summarized below:
>
> **SD v1.4 guidance scale 2.0**
>
> | Metric (IoU / ACC) | TV only | All |
> |--------------------|--------------|--|
> | $\Delta h_{\emptyset}$ |0.813/0.893|0.876/0.911|
> | $\Delta s_{\emptyset}$ |0.827/0.897|0.893/0.920|
> | $\Delta h_{\tilde{\theta}}$ |0.868/0.926|0.858/0.912|
> | $\Delta s_{\tilde{\theta}}$ |0.865/0.923|0.718/0.904|
>
> **SD v2.1 for guidance scale 2.0**
>
> | Metric (IoU / ACC) | TV only | All |
> |--------------------|--------------|--|
> | $\Delta h_{\emptyset}$ |0.930/0.955|0.910/0.961|
> | $\Delta s_{\emptyset}$ |0.925/0.951|0.956/0.976|
> | $\Delta h_{\tilde{\theta}}$ |0.933/0.958|0.888/0.979|
> | $\Delta s_{\tilde{\theta}}$ |0.916/0.944|0.950/0.972|
>
> The performance remains consistent with the results reported in the paper, indicating that our method is not highly sensitive to the guidance scale.
>
> **(2) Baseline checkpoint sensitivity**
>
> We further evaluate robustness to the choice of baseline checkpoint by replacing the original baseline with a more trained model (v1.2) for SD v1.4:
>
> | Metric (IoU / ACC) | TV only | All |
> |--------------------|----------|------|
> | $\Delta h_{\tilde{\theta}}$ | 0.875 / 0.918 | 0.854 / 0.955 |
> | $\Delta s_{\tilde{\theta}}$ | 0.886 / 0.926 | 0.837 / 0.955 |
>
> As discussed in the remark in Section 5.1 and Appendix C, the baseline is not required to be strictly underfitted; it only needs to be relatively less overfitted than the target for the Δ metric to capture incremental curvature.
>
> While our access to intermediate checkpoints is limited, we conjecture that extreme baseline choices (e.g., excessively early checkpoints that fail to capture the general data distribution, or checkpoints too close to the target) may weaken our difference-based methods.
>
> We also note that the unconditional model is the most practical baseline in the widely used classifier-free guidance (CFG) setting.
>
> We will include the additional experiment and discussion in the revision.
>
> > The paper lacks a rigorous execution pipeline diagram, which reduces its reproducibility.
>
> > Please add a detailed execution pipeline diagram for the proposed method.
>
> We thank the reviewer for the helpful suggestion to include an execution pipeline diagram.
>
> We agree that such a diagram would improve clarity and reproducibility, and we will include a detailed pipeline figure as well as a formal algorithmic description in the final version.
>
> > For concept-level memorization, have the authors observed that the method fails systematically, or does it still produce some weaker but interpretable response pattern?
>
> We explored concept-level memorization using prompts such as “Elon Musk is giving a speech” versus “David is giving a speech.” In some cases, the curvature-difference $\Delta h$ signal showed stronger activation around semantically meaningful regions (e.g., facial areas) for specific identities like Elon Musk.
>
> However, this behavior was not consistent across samples. Overall, the method can occasionally produce interpretable signals, but it does not yet exhibit a systematic or robust pattern for concept-level memorization.
>
> We hypothesize that more structured analyses, such as eigenvector analysis or representation-level approaches, may yield more reliable signals.

---

> > ### Author Rebuttal · Reviewer_nvQ6 · 2026-04-04
> >
> > Thanks for clarifying. I will maintain my score.

---

> > > ### Author Response · Authors · 2026-04-06
> > >
> > > We are pleased that our responses have helped clarify your concerns, and we sincerely appreciate your positive feedback on our work.

---

### Official Review · Reviewer_Azy7 · 2026-03-11

**Soundness:** 3
**Presentation:** 3
**Significance:** 3
**Originality:** 3
**Overall Recommendation:** 4
**Confidence:** 3

**Summary:**

This paper addresses the challenge of spatially localizing memorization in diffusion models by proposing an innovative geometric characterization framework. The authors define local memorization as a coordinate-wise variance collapse, which mathematically manifests as high curvature in the model’s learned log-density. To isolate genuine overfitting from intrinsic data constraints, the paper introduces the Curvature-Difference method, which anchors memorized regions by subtracting the curvature of an underfitted baseline model. Furthermore, a first-order score-difference surrogate is derived to alleviate the computational intractability of second-order Hessian calculations.

**Compliance With Llm Reviewing Policy:**

Affirmed.

**Key Questions For Authors:**

1.How sensitive is the localization performance to the baseline model selection? Have you tested scenarios where the baseline model itself is partially overfitted and how does this situation affect the precision-recall trade-off of the Δs metric?

2.Could you explicitly define the algorithmic process for generating these memorization masks? If the masks are obtained from pixel-wise differences between generated and training images, how do you address the false positives in the masks caused by stochastic generation noise or semantic alignments that do not amount to actual memorization?

3.In your experiments, are there specific scenarios such as near-boundary cases or highly complex textures where the first-order Δs metric fails while the second-order Δh metric succeeds when computational resources are sufficient? In other words, does the curvature theory provide predictive insights that the purely empirical score-difference approach fails to capture?

4.What is the specific computational overhead including inference time and peak memory of calculating the Δs metric at each denoising step in comparison with standard inference? For high-resolution models such as SDXL, does the requirement for multiple score evaluations for both the target and baseline models make this method too slow for real-time copyright filtering?

5. We note that this method is relatively general. Can its effectiveness be verified on other generative models? For example, DIT/Flow Matching.

**Limitations:**

Yes

**Strengths And Weaknesses:**

Strengths

1. This paper innovatively introduces the concept of variance collapse from high-dimensional statistical geometry into the privacy research of diffusion models, and provides a rigorous mathematical definition for local memorization by characterizing the sharpness of the log-density landscape with the Hessian matrix, moving beyond heuristic pixel-level observations.

2. The research follows a rigorous logical chain from theoretical definition and method proposal to experimental validation, with standardized use of mathematical notations and intuitive explanations for complex geometric concepts.

3. Departing from the binary "memorized or not" detection in existing studies, this work focuses on the spatial localization of memorized regions, which holds practical value for identifying partial plagiarism in copyright disputes.

Weaknesses

1. The core of the proposed method relies heavily on the selection of underfitted baselines. Improper baseline selection such as slight memorization in baselines or large distribution divergence from target models may introduce massive artifacts or missed detections, yet the paper lacks quantitative guidelines for optimal baseline selection and relevant robustness tests.

2. Defining ground-truth memorization masks for quantitative localization evaluation remains a challenge. If the masks are merely generated via pixel subtraction, the evaluation results may be biased against the density-based geometric method proposed in this paper.

3. Although the score-difference surrogate Δs is presented, the paper provides no detailed comparative tables regarding the additional time overhead and memory consumption caused by calculating Δs in large-scale inference scenarios.

4. The current research only focuses on pixel-level replication, while diffusion models often suffer from semantic memorization (e.g., plagiarism in composition or style with non-identical pixels). The paper fails to provide a clear exploration direction for extending the variance-collapse-based geometric interpretation to latent space or deeper semantic-level localization.

---

> ### Author Rebuttal · Authors · 2026-03-30
>
> Thank you for the valuable feedback. Below, we address the comments.
> > W1, Q1: Sensitivity to baseline selection and effects of partially overfitted baselines
>
> As noted in the remark in Section 5.1, **the baseline model (SD v2.0) is already overfitted enough to show memorization [1]**, but is still less overfitted than the target model (SD v2.1), and still achieves strong performance, as shown in Table 1 and Figure 5. As discussed in Appendix C, the curvature continues to increase even after memorization-like behavior emerges, leading to a measurable curvature gap in memorized regions. Our method detects memorization by capturing this difference.
>
> We further evaluate robustness to baseline choice by replacing the original baseline (v1.1) with a more trained model (v1.2) for SD v1.4.
>
> | Metric(IoU/ACC)| TV only    | All  |
> |----------------|------------|----------|
> | $\Delta h_{\tilde{\theta}}$|0.875/0.918|0.854/0.955|
> | $\Delta s_{\tilde{\theta}}$|0.886/0.926|0.837/0.955|
>
> The results show low sensitivity to the choice of baseline; see Appendix E for results on Realistic Vision v5.1.
>
> > W2: Ground-truth mask construction and bias in evaluation
>
> We agree that the masks in [1] rely on pixel-wise differences between generated and training images. However, regardless of potential bias, this criterion is well aligned with the definition of verbatim memorization, which corresponds to near-exact pixel-level reproduction.
>
> Therefore, pixel-level comparison provides a natural and appropriate way to define memorized regions under the notion of verbatim memorization.
>
> > Q2: How memorization masks are constructed and how false positives are addressed
>
> In our work, we use the ground-truth masks provided in [1] without modification. The memorized prompt dataset introduced in [1] is widely used in the memorization literature.
>
> As described in [1], the mask construction does not follow a single fully specified algorithm, but instead combines several steps, including retrieving the closest training samples, identifying invariant regions via pixel-wise comparisons, and applying thresholding, along with filtering of degenerate cases (e.g., texture-like or background-only matches). While this procedure is heuristic, it helps reduce false positives caused by stochastic noise or loose semantic alignment.
>
> We will clarify these details in the revision. Additional qualitative examples with ground-truth masks are provided in Fig. 4 and 5.
>
> > W3, Q4: Computational overhead of $\Delta s$
>
> We would like to note that $\Delta s$ is evaluated **only at the final denoising step**, rather than at every timestep.
>
> Specifically, for $\Delta s_{\emptyset}$, no additional forward passes are required beyond those already used to generate $x_t$ during standard inference with CFG. The overhead is limited to simple element-wise operations. For $\Delta s_{\tilde{\theta}}$, one additional forward pass is required.
>
> We also note that **Appendix Table 4 reports the inference time** for 50-step sampling (1.08 sec using L40S GPU) as well as the cost associated with different numbers of Hutchinson samples (≈0.03 s/itr).
>
> > W4: Limitations of pixel-level memorization and exploration directions for concept-level memorization
>
> As discussed in the limitations section, we agree that extending our method beyond pixel-level analysis to concept-level analysis is an important direction.
>
> A promising extension would be to incorporate controlled baselines, such as partially masked prompts, and to move beyond coordinate-wise analysis to eigenvector-based or representation-space analysis.
>
> We will include additional discussion on this direction in the final version.
>
> > Q3: Specific scenarios where $\Delta h$ provides advantages over $\Delta s$
>
> Rather than observing advantages of $\Delta h$ in specific scenarios, we find that the difference manifests more consistently overall. In particular, $\Delta s$ effectively suppresses non-memorized regions, but tends to produce noisier activations within memorized regions. Since the link between $\Delta s$ and $\Delta h$ holds only in expectation, $\Delta s$ can exhibit noisier and less spatially coherent activations at the coordinate level, whereas $\Delta h$ yields cleaner and more stable localization overall.
>
> > Q5: Generalization to other generative models
>
> We conducted an experiment by finetuning a pre-trained DiT (facebook/DiT-XL-2-256) on cropped patch-duplicated data to induce memorization. Results are consistent with our main results: https://anonymous.4open.science/r/icml_reply-38EF/DiT.png
>
> Also, note that **Bright Ending [2] is not applicable in this setting due to the absence of cross-attention.**
>
> Since there is currently no established benchmark for memorization in DiT or flow matching models, quantitative evaluation is limited, and we leave it as future work.
>
> [1] Webster, "A Reproducible Extraction of Training Images..." arXiv, 2023.
>
> [2] Chen et al. "Exploring Local Memorization...", ICLR, 2025.

---

> > ### Author Rebuttal · Reviewer_Azy7 · 2026-04-05
> >
> > Maintaining my score.

---

> > > ### Author Response · Authors · 2026-04-06
> > >
> > > We are glad that our responses have successfully addressed your concerns, and we sincerely appreciate your positive feedback on our work.

---

### Decision · Program_Chairs · 2026-04-30

**Decision:**

Accept (regular)

**Comment:**

This paper proposes a method for spatially localizing memorized regions in diffusion models by characterizing local memorization as coordinate-wise variance collapse. After rebuttal, all reviewers gave positive scores. The authors' rebuttal effectively addressed key concerns through baseline sensitivity experiments, clarification that the computational overhead of Δs is minimal, and an additional score-based baseline. Reviewers appreciated the rigorous geometric formulation, the practical value of pixel-level localization, and the model-agnostic design.

The camera-ready should reflect the revisions agreed upon during discussion, including: 1) adding a detailed execution pipeline diagram for reproducibility, 2) clarifying the presentation order of unconditional v.s. less-trained baselines to avoid confusion about the method's practical applicability, and 3) expanding the discussion in Section 4.2 on why the Hessian diagonal is a natural tool for detecting coordinate-wise curvature collapse.